# Conditioning Diffusions Using Malliavin Calculus

**Jakiw Pidstrigach** [* 1]  **Elizabeth L. Baker** [* 2]  **Carles Domingo-Enrich** [3]  **George Deligiannidis** [1]  **Nikolas Nüsken** [* 4]

## Abstract

In generative modelling and stochastic optimal control, a central computational task is to modify a reference diffusion process to maximise a given terminal-time reward. Most existing methods require this reward to be differentiable, using gradients to steer the diffusion towards favourable outcomes. However, in many practical settings, like diffusion bridges, the reward is singular, taking an infinite value if the target is hit and zero otherwise. We introduce a novel framework, based on Malliavin calculus and centred around a generalisation of the Tweedie score formula to nonlinear stochastic differential equations, that enables the development of methods robust to such singularities. This allows our approach to handle a broad range of applications, like diffusion bridges, or adding conditional controls to an already trained diffusion model. We demonstrate that our approach offers stable and reliable training, outperforming existing techniques. As a byproduct, we also introduce a novel score matching objective. Our loss functions are formulated such that they could readily be extended to manifold-valued and infinite dimensional diffusions.

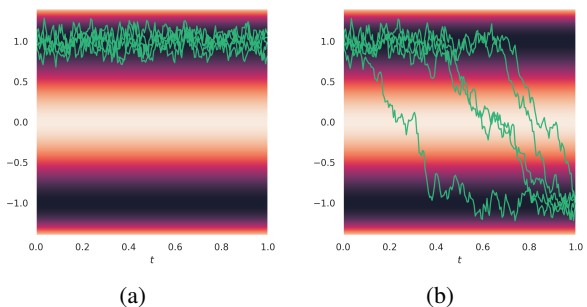

(a)          (b)

Figure 1: The figure depicts particle trajectories in a double well potential, with the background color indicating potential intensity. The potential has two metastable states at $x = 1$ and $x = -1$. In (a) we observe that under the diffusion dynamics, particles initialised at $x = 1$ typically remain confined to their well, rarely crossing the barrier to $x = -1$. On the right, in (b), the diffusion is conditioned on the rare event of transitioning between the two metastable states.

and generative modelling (Dhariwal & Nichol, 2021; Ho & Salimans, 2021), including guidance for generative models (Zhang et al., 2023; Denker et al., 2024).

## 1. Introduction

Simulating conditioned diffusions is a central computational task in many applications, ranging from molecular dynamics and physical chemistry (Dellago et al., 2002; Bolhuis et al., 2002; Vanden-Eijnden et al., 2010) and genetics (Wang et al., 2011) to finance, econometrics (Bladt & Sørensen, 2014; Elerian et al., 2001; Durham & Gallant, 2002), evolutionary biology (Arnaudon et al., 2023; 2017; Baker et al., 2024),

---

[*]Equal contribution [1]Department of Statistics, University of Oxford, UK [2]Department of Computer Science, University of Copenhagen, Denmark [3]Microsoft Research New England, Cambridge, USA [4]Department of Mathematics, King's College, London, UK. Correspondence to: Jakiw Pidstrigach <mail@jakiw.com>, Elizabeth L. Baker <eloba@dtu.dk>.

*Proceedings of the 42$^{nd}$ International Conference on Machine Learning*, Vancouver, Canada. PMLR 267, 2025. Copyright 2025 by the author(s).

**Reference System.** To demonstrate some of the core ideas, let us consider a diffusion process of the form

$$\mathrm{d}X_t = b(X_t)\,\mathrm{d}t + \mathrm{d}B_t, \tag{1}$$

where the drift vector field $b$ is given–either from a learned generative model or from physically or financially motivated considerations. Samples from (1) can be obtained by straightforward numerical simulation.

**Guidance and Control.** In many applications, it is desirable to condition the reference process (1). For instance, in generative modeling, a large pretrained model may not offer the necessary task-specific controls. One may then seek to impose such controls post hoc—for example, to generate images matching a desired edge map or human pose, or to sample proteins with properties tailored to a particular application. Similarly, when the reference process models weather dynamics, interest may lie in rare or extreme

scenarios rather than typical ones.

**Bayesian Inverse Problems.** These objectives can be formalised within the framework of Bayesian inverse problems (Stuart, 2010). To this end, let $G$ be an observation operator and $Y$ the corresponding observation at terminal time:

$$Y = G(X_T). \qquad (2)$$

The goal is to sample $(X_t)_{0 \leq t \leq T}$ given $Y = y$. This setting encompasses all previous examples; for instance, $G$ may extract edge or pose information from an image. Our formulation allows $G$ to be a full or partial (potentially noisy) observation of $X_T$. Note that $G$ may be stochastic; for example, $G(X_T) = X_T + \xi$, where $\xi$ is a random variable.

**Stochastic Optimal Control and Conditioning Diffusions.** In order to condition on $Y$, we add a control term $u_t$ to the reference process (1):

$$\mathrm{d}X_t = b(X_t)\,\mathrm{d}t + u_t(X_t)\,\mathrm{d}t + \mathrm{d}B_t. \qquad (3)$$

The optimal control will have the property that each sample $(X_t)_{0 \leq t \leq T}$ from (3) is a sample from the distribution of (1), conditioned on $Y = y$.

**Likelihoods and Rewards with Gradients.** The function $G$ induces a likelihood or reward

$$g(x; y) := p(Y = y \mid X_T = x). \qquad (4)$$

Current methods, framed in terms of stochastic optimal control (Domingo-Enrich, 2024; Zhang & Chen, 2022; Berner et al., 2024), critically rely on gradient information $\nabla g$ or $\nabla \log g$ to guide the process and learn $u_t$.

**Singular Rewards.** In many settings there is no gradient information available for $g$. One case where this happens is if $G$ is an indicator function or discontinuous, as would be the case for classification. But even if $G$ is smooth but deterministic, the induced likelihood $g$ is often singular if one does not add artificial noise to the observations. Even in the seemingly straightforward case of $G = \mathrm{Id}$, the reward would be given by a Dirac delta distribution $g(x_T; y) = \delta_y(x_T)$, which is not even continuous. The case of $G = \mathrm{Id}$ conditions diffusions to end at a specific state and is known under the term *diffusion bridges* (see Figure 1(a)). Due to $g$ being a Dirac it is in some sense the most challenging setting and will be a guiding problem for us in the development of the methodology.

**Integration by Parts on Path Space.** In this paper, we circumvent these issues altogether by constructing numerical methods that remain unaffected by singularities in the likelihood $g$. The key idea is best introduced via an analogy: replace, for the moment, the trajectory space associated with (1) by the real line. The classical *integration-by-parts* identity

$$\int_{-\infty}^{\infty} \partial_x g(x) f(x)\,\mathrm{d}x = -\int_{-\infty}^{\infty} g(x) \partial_x f(x)\,\mathrm{d}x, \qquad (5)$$

valid provided $f$ and $g$ vanish sufficiently fast at infinity, offers a blueprint for handling singularities: the left-hand side can be given a rigorous meaning even if $g$ lacks differentiability, as long as $f$ is smooth. Numerically, large (exploding) gradients can be avoided by shifting differentiation onto the factor with better properties.

**Outline and Contributions.** Guided by (5), we develop loss functions $\mathcal{L}(u)$ whose unique minimisers $u_t(x; y)$ implement the conditioning of (1) via the controlled diffusion in (3), for *any* condition $Y = y$ at once. Towards this goal,

- we recall the well-known (Eberle, 2015; Denker et al., 2024; Shi et al., 2024; Du et al., 2024) connection of diffusion bridges to Doob's $h$-transform in Section 2.1, in particular the relevance of conditional score functions,

- lift (5) to integration by parts on the space of trajectories; the role of the Lebesgue measure $\mathrm{d}x$ is replaced by the law of $(X_t)_{0 \leq t \leq T}$, and the ordinary derivative $\partial_x$ is replaced by the Malliavin derivative (Nualart, 2006). Based on this, we derive a novel formula for conditional scores that generalises Tweedie's formula (Efron, 2011) for denoising score matching (Vincent, 2011),

- discuss implementation details and showcase numerical performance in Section 4.

Furthermore, we recover existing methods for stochastic optimal control from our framework, see Section 3.2.

## 2. Theoretical Background and Main Result

Theorem 2.1 below serves as the mathematical foundation of our methodology. Before presenting it, we introduce the necessary notation and assumptions.

Throughout, we consider diffusion processes of the form

$$\mathrm{d}X_t = b_t(X_t)\,\mathrm{d}t + \sigma_t(X_t)\,\mathrm{d}B_t, \qquad X_0 = x_0, \quad (6)$$

where $b : [0, T] \times \mathbb{R}^n \to \mathbb{R}^n$ is a smooth drift of at most linear growth, $x_0 \in \mathbb{R}^n$ is a fixed initial state, $B_t$ is a standard Brownian motion, and $\sigma : [0, T] \times \mathbb{R}^n \to \mathbb{R}^{n \times n}$ specifies the volatility, assumed to be symmetric, strictly positive definite and bounded, with bounded inverse. A key component is the *Jacobian* $J_{t|s}$ associated to (6), which is a matrix-valued stochastic process satisfying

$$\mathrm{d}J_{t|s} = \nabla b_t(X_t) J_{t|s}\,\mathrm{d}t + \nabla \sigma_t(X_t) J_{t|s}\,\mathrm{d}B_t, \quad (7)$$

with initial condition $J_{s|s} = \mathrm{Id}$, for fixed $s \in [0, T]$. Intuitively, $J_{t|s}$ measures the sensitivity of (6) at time $t$ with respect to a small perturbation at an earlier time $s < t$. More precisely, it represents the derivative process, i.e.

$J_{t|s} = \nabla_{X_s} X_t$ (Williams & Rogers, 1979, Chapter V.13). The process $J_{t|s}$ plays a central role in adjoint methods for gradient computation (Li et al., 2020) and stochastic optimal control (Domingo-Enrich et al., 2024b), and we highlight the fact that simulating the full matrix-valued evolution is typically unnecessary as only matrix-vector products are required (see Appendix A). Our main theoretical contribution can now be stated as follows.

**Theorem 2.1.** *Let $\alpha : [0, T] \to \mathbb{R}^{n \times n}$ be a matrix-valued differentiable function such that $A_{T|s} := \alpha_T - \alpha_s$ is invertible for all $s \in [0, T)$. Define the* score process

$$\mathcal{S}_s := A_{T|s}^{-1} \int_s^T \alpha_t' J_{t|s}^\top (\sigma_t(X_t)^\top)^{-1} \, \mathrm{d}B_t, \qquad (8)$$

*as well as the loss functional*

$$\mathcal{L}(u) = \mathbb{E}\left[ \int_0^T \|u_s(X_s; Y) - \mathcal{S}_s\|^2 \, \mathrm{d}s \right], \qquad (9)$$

*where the expectation is taken with respect to the injected noise $(B_t)_{0 \leq t \leq T}$ driving (6), (7) and (8), as well as potential noise in $G$ in (2). Then $\mathcal{L}$ admits a unique minimiser $u^*$, and for any $y \in \mathbb{R}^d$, the law of*

$$\begin{aligned}
\mathrm{d}X_t &= b_t(X_t) \, \mathrm{d}t + \sigma_t(X_t)\sigma_t(X_t)^\top \, u_t^*(X_t; y) \, \mathrm{d}t \\
&\quad + \sigma_t(X_t) \, \mathrm{d}B_t
\end{aligned} \qquad (10)$$

*coincides with the conditional law of (6), given $Y = y$.*

Based on Theorem 2.1, we propose to (i) estimate the expectation in (9) using Monte Carlo, (ii) parameterise the drift $u_t$ (including the conditioning) by a neural network, and (iii) learn the parameters of $u_t$ through gradient-descent type updates. We have formalised the resulting training procedure in Algorithm 1, and, given the close relationship between Theorem 2.1 and the Bismuth-Elworthy-Li (BEL) formula from Malliavin calculus (see Section 2.2), we refer to these methods as *BEL-algorithms*. Importantly, the score process (8) can be simulated efficiently by directly updating $J_{t|s}^\top(\sigma_t(X_t)^\top)^{-1} \, \mathrm{d}B_t$ along (7), without simulating the full dynamics of $J_{t|s}$ (see Appendix A for details). The main hyperparameter in Algorithm 1 is the matrix-valued function $\alpha : [0, T] \to \mathbb{R}^{n \times n}$; based on variance and stability considerations we give guidance on its choice in Section 3. Furthermore, specific choices of $\alpha$ allow us to connect existing methods to the framework of Theorem 2.1; those are explained in Sections 3.2.1 and 3.2.2.

*Remark* 2.2 (Amortisation). After training according to Algorithm 1, the learned control $u_t(x; y)$ performs conditioning of (1) for arbitrary $y$, without the need for retraining. On a related note, the constructions in (8) and (9) as well as in Algorithm 1 depend on the observation operator $G$ only implicitly through the samples $Y = G(X_T)$ and do not access or require any particular structure; in particular, they

are gradient free. On the technical technical level this corresponds to the conditional expectation being the minimiser of the $L^2$ loss, see (17).

In the remainder of this section we prove Theorem 2.1 and illustrate its connection to integration by parts, as hinted at in (5). Implementation details are deferred to Section 4.

---

**Algorithm 1** BEL - Training Step

---

**Require:** $\alpha : [0, 1] \to \mathbb{R}^{n \times n}$, initial condition $x_0$, batch size $N$, current drift approximation $u^\theta$, time grid $\{t_0, t_1, \ldots t_M\}$.
1: **for** $i = 1$ to $N$ **do**
2:     Sample a sample path $X$ with corresponding Brownian motion path $B$ from the SDE (6).
3:     Sample an observation $Y = G(X_T)$ from (2).
4:     Compute the Monte Carlo estimator for $\mathcal{S}_s$ using (8) (for details see Algorithm 2) along the path $(X, B)$.
5:     Calculate the single-path loss

$$l_i(\theta) = \sum_{j=1}^{M-1} \|u_{t_j}^\theta(X_{t_j}; Y) - \mathcal{S}_{t_j}(X, B)\|^2,$$

6: **end for**
7: Sum for the full-batch loss

$$\mathcal{L}^N(\theta) = \sum_{i=1}^N l_i(\theta).$$

8: Take a gradient step on $\mathcal{L}^N(\theta)$ with your favourite optimiser.

---

### 2.1. Doob's $h$-transform and Conditional Scores

As a first step towards Theorem 2.1, we recall the fact that the optimal drift in (10) can be expressed in terms of conditional scores (Rogers & Williams, 2000, p. 83):

**Proposition 2.3** (Doob's $h$-transform). *For $y \in \mathbb{R}^d$, let*

$$u_t^*(x; y) := \sigma_t(x)\sigma_t(x)^\top \nabla_x \log p(Y = y \mid X_t = x), \qquad (11)$$

*where $p(Y = y \mid X_t = x)$ is the probability of $Y = y$ given $X_t = x$. Then the corresponding controlled diffusion (10) reproduces the conditional law of (6), given $Y = y$.*

Similar closed-form formulations are available in the context of optimal control (Nüsken & Richter, 2021, Section 2.2) and time reversals (Boffi & Vanden-Eijnden, 2024; Song et al., 2021b; Ho et al., 2020). Note that for $t = T$, the conditional score coincides with the gradient of the log-reward, $\nabla_x \log p(Y = y \mid X_T = x) = \nabla_x \log g(x; y)$. As a consequence, if $g$ is singular, (11) becomes singular (and possibly numerically unstable) as $t \to T$. We deal with this issue in the next subsection.

## 2.2. Malliavin Calculus and Integration by Parts

While Proposition 2.3 in principle identifies the desired control vector field, the right-hand side of (11) will be unavailable in all but simple toy examples. The following result provides a formula that is amenable to Monte Carlo simulation:

**Proposition 2.4** (Generalised Tweedie formula). *The conditional score is given by the conditional expectation of the score process,*

$$
\begin{aligned}
\nabla_x \log p(Y = y \mid X_t = x) \\
= \mathbb{E}\left[\mathcal{S}_t \mid X_t = x, Y = y\right],
\end{aligned} \tag{12}
$$

*for all $t \in [0, T]$ and $x \in \mathbb{R}^n, y \in \mathbb{R}^d$.*

*Remark* 2.5. In the case when (6) is linear, Proposition 2.4 simplifies to the celebrated Tweedie formula (Efron, 2011), given, e.g., by

$$
\nabla_{x_T} \log p_T(X_T = x_T) = \tfrac{1}{T}(\mathbb{E}[X_t \mid X_T = x_T] - x_T).
$$

Similarly, the *denoising score matching objective* for diffusion models (Song et al., 2021a) can be seen as an instance of (9) for a specific choice of $\alpha$. Consequently, (12) is a generalisation to nonlinear SDEs, also allowing us to derive novel regression targets for diffusion models; see Appendix B.

*Proof sketch (Proposition 2.4).* See Appendix D.1 for full details. Without loss of generality, we may consider the diffusion bridge setting (i.e., $G(X_T) = X_T$), see Lemma D.2. To show (12), we rely on two main ideas:

Firstly, to express the transition probability in terms of an expectation, we use the fact that whenever a random variable $X$ admits a smooth Lebesgue density $p_X$, we have the representation

$$
p_X(x) = \mathbb{E}[\delta_x(X)], \tag{13}
$$

see Duistermaat et al. (2010) for a rigorous general account or Watanabe (1987, Section 2.1) for the statement in the context of Malliavin calculus.

Secondly, as outlined in the introduction, we elevate the integration-by-parts formula (5) to Wiener space, i.e., to the space of sample paths of Brownian motion $(B_t)_{0 \le t \le T}$. To this end, we introduce the Malliavin derivative $D_t$, which represents differentiation with respect to the infinitesimal noise increment $\mathrm{d}B_t$: for a functional $g$ that depends on the realisation of $(B_t)_{0 \le t \le T}$, the Malliavin derivative is given by

$$
D_t g = \frac{\partial g}{\partial \, \mathrm{d}B_t}. \tag{14}
$$

A rigorous treatment of (14) requires the framework of calculus on Wiener space; see Nualart (2006). With (14) in

place, the analogue of (5) on Wiener space becomes

$$
\mathbb{E}\left[\int_0^T D_t g \cdot f_t \, \mathrm{d}t\right] = \mathbb{E}\left[g \int_0^T f_t \cdot \mathrm{d}B_t\right], \tag{15}
$$

for appropriate choices of $g$ and the stochastic process $f_t$.

Using (13), we write

$$
\nabla_x \log p_{T|t}(X_T = x_T \mid X_t = x) = \frac{\nabla_x \mathbb{E}[\delta_{x_T}(X_T) \mid X_t = x]}{p_{T|t}(X_T = x_T \mid X_t = x)},
$$

and, using the chain rule, we further obtain

$$
\begin{aligned}
\nabla_x \mathbb{E}[\delta_{x_T}(X_T) \mid X_t = x] \\
= -\mathbb{E}[\nabla_{x_T} \delta_{x_T}(X_T) J_{T|t}^\top \mid X_t = x].
\end{aligned}
$$

Here, $\nabla_{X_t} X_T = J_{T|t}$, as explained in Section 2, and the gradient $\nabla_{x_T} \delta_{x_T}$ is understood in the sense of distributions (Duistermaat et al., 2010, Chapter 4). To complete the proof, it remains to eliminate the derivative on $\delta_{x_T}$ via the integration by parts formula (15)—the left-hand side of (15) precisely explains the appearance of the stochastic integral in the score process (8). For this, we need to convert the conventional gradient $\nabla_{x_T}$ into the Malliavin derivative $D_t$. This is achieved using the matrix-valued function $\alpha$, as detailed in Appendix D.1. In fact, there are infinitely many ways to carry out this conversion, each determined by a particular choice of $\alpha$. We note that the proof strategy, particularly the conversion of $\nabla_{x_T}$ into $D_t$ and the use of integration by parts, closely resembles the proof of the Bismuth-Elworthy-Li (BEL) formula from Malliavin calculus (Bismut, 1984; Elworthy & Li, 1994), which inspired the name of Algorithm 1. $\qquad \square$

## 2.3. Amortised Bridges and Forward-KL

While Propositions 2.3 and 2.4 together provide an expression for the desired control vector field $u_t^*$, the left-hand side of (12) requires access to the targeted distribution of (6), conditioned on $X_t = x$ and $Y = y$; the construction might thus appear circular. Fortunately, taking expectations over $X_t$ and $Y$ in (12) not only recovers the straightforward-to-simulate reference diffusion (6) as a mixture of its bridges, but also allows us to amortise the learning procedure, inferring all the bridges simultaneously.

The following proof of our main result reflects this idea:

*Proof of Theorem 2.1.* The minimiser of the least-squares "local-in-time" loss

$$
\mathcal{L}_{\mathrm{local}}^t(u) := \mathbb{E}\left[\|u_t(X_t, Y) - \mathcal{S}_t\|^2\right] \tag{16}
$$

is given by the conditional expectation of $\mathcal{S}_t$ with respect to $X_t$ and $Y$, i.e.

$$
u_t^*(x, y) = \mathbb{E}[\mathcal{S}_t \mid X_t = x, Y = y]. \tag{17}
$$

Combining Proposition 2.3 and Proposition 2.4, this implies that the minimiser $u_t^*$ is equal to the optimal control term. By exchanging the expectation with the integral in (9), we get that $u_t$ is equal to the optimal control $\nabla_{X_t} \log p(Y \mid X_t)$, for each $t \in [0, T]$.

$\square$

*Remark* 2.6 (KL interpretation). The time-integrated local loss can be interpreted in terms of the amortized KL-divergence between measures on path space,

$$\int_0^T \mathcal{L}_{\text{local}}^t(u_t)\, \mathrm{d}t = \mathbb{E}\left[\mathrm{KL}(\mathbb{P}^y \mid \mathbb{P}^u)\right] + C \qquad (18)$$

with a constant $C > 0$ that does not depend on $u$. Here, the expectation is taken with respect to $Y$ and $\mathbb{P}^y$ refers to the distribution of trajectories induced by the diffusion conditioned on $Y = y$. The measure $\mathbb{P}^u$ refers to the distribution associated to the current control $u$. The forward-KL divergence in (18) is mode-covering (Naesseth et al., 2020), which is a desirable property for conditional generation and rare event simulation. Furthermore, the KL-representation (18) allows us to connect the BEL-framework from Algorithm 1 to the previous works by Heng et al. (2022) and Baker et al. (2025). However, the former requires additional numerical overhead in learning backward processes, whereas the latter relies on additional simulations for a Feynman-Kac type construction. The accuracy of these approaches is evaluated experimentally in Section 4. Finally, we remark that (18) distinguishes our approach from the work by Du et al. (2024), who use forward-KL, together with subsequent (Gaussian) approximations. For background on the "measures on path space" perspective, see Nüsken & Richter (2021), and for a proof of (18) see Appendix D.2.

## 3. On the Choice of $\alpha$

Theorem 2.1 provides a whole class of loss functions for diffusion bridge simulation: each choice of $\alpha$ yields a slightly different approach to approximating the drift in (11), providing considerable flexibility. We now exploit this flexibility for two purposes: in Section 3.1, we derive an optimal choice of $\alpha$ under a simplified setting, and in Section 3.2, we connect our bridge losses to existing algorithms used in stochastic optimal control and diffusion bridge simulation.

### 3.1. Optimal $\alpha$ for Reduced Variance

Each choice of $\alpha$ in Theorem 2.1 implies a different regression target for the neural network. While all of these targets have the same mean, namely the diffusion bridge drift in (11), they differ in variance:

$$\mathrm{Var}(\mathcal{S}_s) = \mathrm{Var}\left( A_{T|s}^{-1} \int_s^T \alpha_t' J_{t|s}^\top (\sigma_t(X_t)^\top)^{-1} \mathrm{d}B_t \right).$$

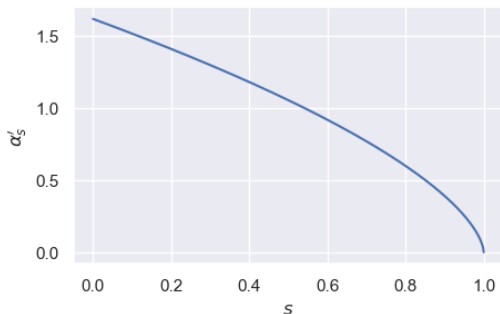

Figure 2: The variance-optimal weighting $\alpha_s'$ (see Lemma 3.1) for the simplified setting in which $X_t$ is a conditioned Brownian motion.

Since the algorithm works by generating independent samples from $\mathcal{S}_s$ and then regressing against those, a lower variance of $\mathcal{S}_s$ is expected to lower the variance of the gradients. In the following result we compute and optimise the variance in a simplified setting, providing guidance for the choice of $\alpha$:

**Lemma 3.1.** *Set $Y = X_T$, $T = 1$, $n = 1$, $b = 0$ and $\sigma = 1$, i.e., $X_t$ is a one-dimensional Brownian motion conditioned on $X_0 = x_0$ and $X_1 = x_1$. Then the variance of the Monte-Carlo estimator*

$$\nabla \log p_{1|0}(B_1 = x \mid B_0 = x_0) \approx \int_0^1 \alpha_t' J_{t|0}^\top (\sigma_t^{-1})^\top \mathrm{d}B_t$$

*is given by*

$$\frac{1}{\alpha_1 - \alpha_0}\left( \int_0^1 (\alpha_t')^2\, \mathrm{d}t + \int_0^1 \frac{(\alpha_1 - \alpha_t)^2}{(1-t)^2}\, \mathrm{d}t + d^2 \right), \quad (19)$$

*assuming that $\frac{\alpha_1 - \alpha_s}{\sqrt{1-s}} \to 0$ as $s \to 1$ and letting $d = x_1 - x_0$. The variance is minimised for the choice*

$$\alpha_t' = 1 - (1-t)^{\frac{1}{2}(1+\sqrt{5})}. \qquad (20)$$

The proof of Lemma 3.1 can be found in Appendix E, and Figure 2 shows the derivative $\alpha_t'$ of the optimal variance-reducing choice in (20). Interpreted as weights in the score process (8), we see that (20) weights the initial increments of the Brownian motion relatively highly in comparison to increments closer to the terminal time. The choice of $\alpha$ will further be explored in Section 4.

### 3.2. Connections to other Algorithms

#### 3.2.1. REPARAMETRISATION

Inspired by the reparametrisation trick in variational inference (Kingma et al., 2019, Section 2.4), Domingo-Enrich et al. (2024b) derived a novel methodology for stochastic optimal control. A variant of it–adapted to our setting–can be recovered from Theorem 2.1 by a specific choice of $\alpha$:

**Lemma 3.2.** *Let $M : [0, T] \to \mathbb{R}^{n \times n}$ be a matrix-valued differentiable function such that $M_0 = \text{Id}$ and $M_T = 0$. Then the score process from (8) has the equivalent representation*

$$\mathcal{S}_s = \int_s^T (M_t \nabla b_t^\top(X_t) - M_t')(\sigma_t(X_t)^\top)^{-1} \, dB_t. \quad (21)$$

*Proof.* The correspondence between (8) is via the choice $\alpha_t = J_{t|s}^{-1} M_t$; see Appendix D.3. $\square$

The significance of Lemma 3.2 is that (i) it extends the reparameterisation trick to the singular reward setting and (ii) it gives conceptual insights into the choices of $\alpha_t$ and $M_t$ in the respective methods. For further intuition into the role of $M_t$ we refer to Domingo-Enrich et al. (2024b) and Appendix D.3.

### 3.2.2. GAUSSIAN APPROXIMATIONS

Another method to learn $\nabla \log p(Y = y \mid X_s = x_s)$, identified as the conditional drift in Proposition 2.4, is to approximate the transition densities as Gaussian and regress against their score. We now derive methods based on a Gaussian approximation and show how they can be reformulated as BEL algorithms for specific choices of $\alpha$. Moreover, the BEL algorithm provides deeper insight into the approximation error introduced by the Gaussian assumption.

Rather than regressing directly against the target, we first observe that one can regress against the optimal drift for a small timestep:

**Lemma 3.3.** *For any $t \geq s$,*

$$
\begin{aligned}
&\mathcal{L}^{GA}(u, s, t) \\
&:= \mathbb{E}\big[\|u_s(X_s, y) - \nabla_{X_s} \log p_{t|s}(X_t \mid X_s)\|^2\big]
\end{aligned}
\quad (22)
$$

*is minimised by the diffusion bridge term $u_s^*(x_s, y)$, see (11).*

Thus, optimising $\mathcal{L}^{GA}$ for each $s$ yields the diffusion bridge drift. To achieve this, one must select a $t \geq s$ for each $s$. After discretising the time domain into $\{t_0, t_1 = t_0 + \delta t, \ldots, t_N = t_0 + N\delta t\}$, a suitable loss function is:

$$\mathcal{L}^{GA}(u) = \sum_{i=1}^{N-1} \mathcal{L}^{GA}(u, t_i, t_i + \delta t). \quad (23)$$

For small $\delta t$, the transition density can be approximated by a Gaussian, for example, via an Euler-Maruyama step:

$$p(X_{t+\delta t} \mid X_t) \approx \mathcal{N}(X_t + \delta t \, b_t(X_t), \delta t \, a_t(X_t)), \quad (24a)$$

$$a_t(X_t) = \sigma_t(X_t)\sigma_t(X_t)^\top. \quad (24b)$$

This provides an explicit expression for $\nabla_{X_t} \log p_{t+\delta t|t}(X_{t+\delta t} \mid X_t)$, which can be used for regression. In Heng et al. (2022), the authors applied a similar Gaussian approximation to the time-reversal of a diffusion to learn bridges of the time-reversed process.

However, this approximation is performed in the density domain $p_{t+\delta t|t}(X_{t+\delta t} \mid X_t)$, and its impact on the accuracy of $\nabla \log p_{t+\delta t|t}(X_{t+\delta t} \mid X_t)$ is unclear. By the generalised Tweedie formula (see Proposition 2.4), we obtain the explicit relation:

$$\nabla \log p_{t+\delta t|t}(X_{t+\delta t} \mid X_t) = \mathbb{E}[\mathcal{S}_s \mid X_t, X_{t+\delta t}], \quad (25)$$

where $\mathcal{S}_s$ is defined in (8) with terminal time $T = t + \delta t$. The Gaussian approximation can now be directly related to a discretisation scheme for $\mathcal{S}_s$ in (25). We prove this in the case of $\sigma = 1$:

**Lemma 3.4.** *Approximating $p(X_{t+\delta t} \mid X_t)$ by a Gaussian (24a) and regressing against its score is equivalent to choosing $\alpha_s' = 1_{[t, t+\delta t]}$ and approximating the stochastic integral in $\mathcal{S}_s$ (8) as*

$$\int_t^{t+\delta t} J_{s|t} dB_s \approx J_{t+\delta t|t}(B_{t+\delta t} - B_t), \quad (26)$$

*and furthermore approximating $J_{t+\delta t|t}$ by an Euler-Maryuama step on (7):*

$$J_{t+\delta t|t} \approx \text{Id} + \delta t \nabla b(t, X_t). \quad (27)$$

The proofs for this section can be found in Appendix D.4.

### 3.3. Summary of Choices

Based on the above discussion we will now summarise some choices for the function $\alpha_t$ in Theorem 2.1. Each choice leads to a different loss function, and we compare the different choices empirically in Section 4.

**BEL optimal:** The first choice is setting $\alpha_s$ as in Lemma 3.1. Although this only gives optimal variance in the specific case of Brownian motion, it still may do well in other problem settings, especially those similar to Brownian motions.

**BEL first:** Based on Lemma 3.3 we can set $\alpha_s$ such that $\alpha_s' = 1_{[s, s+\Delta s]}$. In this case, $\alpha_T - \alpha_s = \Delta s$. This means we are only using local information to approximate the score.

**BEL average:** Another choice is setting $\alpha_s = s$. This leads to the traditional Bismut-Elworth-Li formula (Bismut, 1984; Elworthy & Li, 1994).

**BEL last:** This algorithm uses $\alpha_s' = 1_{[T-\Delta t, T]}$. This is similar to the stochastic optimal control setting, where the gradient of the target function is propagated back through the whole trajectory.

**Reparametrisation:** Corresponding to the discussion on the reparametrisation trick in Lemma 3.2, we set $\alpha_s = J_{s|t}M_s$, with $M_s = \frac{T-s}{T} \text{Id}$.

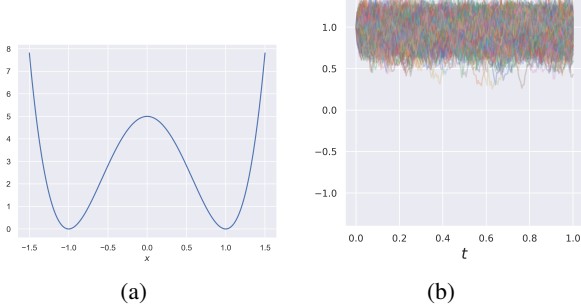

(a)       (b)

Figure 3: In (a), we plot the double-well potential (31) for $v = 5$. In (b), we sample 1,000 paths from the unconditioned SDE (30) and observe that they remain confined to a single potential minimum, failing to transition between them. This highlights the inherent difficulty of the problem.

# 4. Experiments

In this section, we do an empirical evaluation of our methods. Full experimental details are provided in Appendix C. In Section 4.1 and Section 4.2 we study the case of diffusion bridges, i.e. $Y = X_T$, or a Dirac-delta reward function $g$, since this can be seen as the most challenging setup: In Section 4.1, we conduct experiments where the true transition densities are available for evaluation. In Section 4.2, we demonstrate our method on bridges of stochastic shape processes, which have applications in biology. We also compare to related methods (Heng et al., 2022) and (Baker et al., 2025) and show favourable results. In Section 4.3 we apply our methodology to diffusion models or equivalently, flow matching algorithms.

## 4.1. Controlled Environment Experiments

In this subsection, we introduce two experiments where the true diffusion bridge drift can be computed, allowing us to evaluate our methods against the ground truth. We explain the experiment setup in Appendix C.1.1.

### 4.1.1. METHODOLOGY

For the experiments in Section 4.1.2 and Section 4.1.3, we consider one-dimensional SDEs, simulated independently across all dimensions. Trajectories are conditioned to start at $y_{\text{init}} = (1, 1, \dots, 1) \in \mathbb{R}^D$ and end at $y_{\text{final}} = (-1, 1, \dots, 1) \in \mathbb{R}^D$, modifying the first coordinate.

This setup isolates the effect of the rare event from the dimensionality of the state space. Conditioning all coordinates would cause the probability of the transition event—being the product of independent one-dimensional transition probabilities—to scale as $p^D$, where $p$ is the probability of the event in one dimension. In contrast, by constraining only

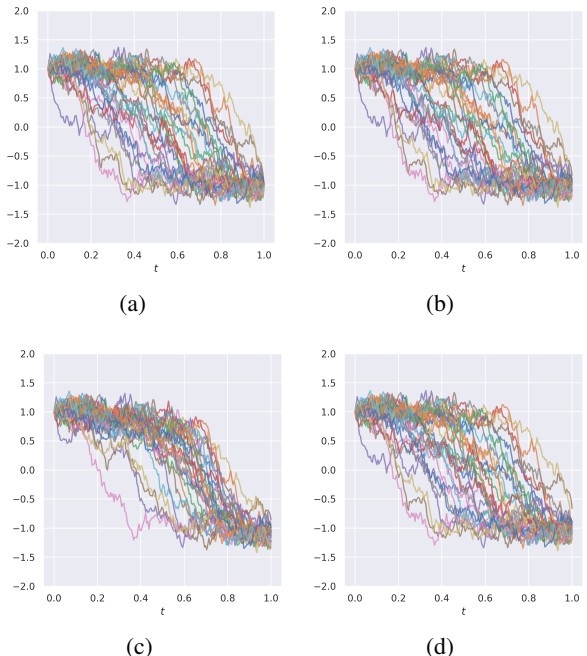

(a)       (b)

(c)       (d)

Figure 4: Panel (a) shows ground truth paths from (30), conditioned on transitioning from state 1 to state $-1$. Panels (b), (c) and (d) present the paths generated by the BEL-first, BEL-last and BEL-optimal respectively.

the first coordinate, we maintain an approximately constant event probability whilst varying the problem dimensionality.

### 4.1.2. BROWNIAN MOTION

The first experiment we consider involves conditioning a Brownian motion:

$$\mathrm{d}X_t = \mathrm{d}B_t. \qquad (28)$$

For this SDE, the diffusion bridge drift (11) has a closed-form solution given by:

$$\nabla \log p_{1|t}(X_1 = x \mid X_t = x_t) = -\frac{x_t - x}{1 - t}. \qquad (29)$$

This allows us to compare our methods against the true bridges. The results are presented in Table 4. We observe that the reparametrisation trick algorithm performs well for the one-dimensional Brownian motion, while the BEL average achieves the best performance in 10 dimensions. For an explanation of the metrics see Appendix C.1.2

### 4.1.3. DOUBLE-WELL

In this experiment, we consider the double-well problem as described by Nüsken & Richter (2021). This model is given

by the SDE:

$$dX_t = -\nabla U_v(X_t)dt + dB_t, \qquad (30)$$

$$U_v(x) = v(x^2 - 1)^2. \qquad (31)$$

For $v = 5$, we visualize the potential in Figure 3(a). The potential exhibits two minima at $x = -1$ and $x = 1$, which correspond to *metastable* states. Since the drift term in (30) drives trajectories toward the minima of $U_v$, transitions between $x = -1$ and $x = 1$ are rare events, with their probability decreasing exponentially as the potential barrier height $v$ increases (Kramers, 1940; Berglund, 2013). We illustrate this by plotting 1000 sample paths of the unconditioned process in Figure 3(b), none of which crosses the barrier.

To obtain a ground truth for this example, we numerically estimate $f(x) = p_{1|s}(X_1 = -1 \mid X_s = x)$ and compute its logarithmic gradient. We then compare the paths of the process under the true drift with our estimates in Figure 4.

Additionally, we extend the experiment to higher-dimensional settings as described in Section 4.1.1. The first and second marginals of a 10 dimensional process are shown in Figure 5.

Finally, we quantitatively compare the performance of different algorithms in Table 7. We observe that BEL Last and the Reparameterisation Trick exhibit the lowest performance, aligning with the empirical experience of the authors. For an explanation of the metrics, see Appendix C.1.2. In Figure 7 we also provide plots for the reparametrisation trick.

## 4.2. Shape Processes

We demonstrate our methodology on stochastic shape processes (Arnaudon et al., 2023), which are used in computational anatomy to model morphological changes in human organs due to disease (Arnaudon et al., 2017). In evolutionary biology, they help analyse morphometric changes in species, such as how butterfly wing shapes evolve along phylogenetic trees (Baker et al., 2024).

Following the setup in (Sommer et al., 2021), we consider a shape represented by $x_0 \in \mathbb{R}^{2N}$, where $N$ points discretise a two-dimensional shape. Let $\{y^j\}_{j=1}^M \subset \mathbb{R}^2$ be equidistant points, and $\{B^j\}_{j=1}^M$ be independent Brownian motions in $\mathbb{R}^2$. The SDE governing the shape evolution is given by

$$dx_t^i = \sum_{j=1}^M k(y^j, x_t^i), dB_t^j, \quad 1 \le i \le N, \qquad (32)$$

$$k(x, y) = \kappa \frac{\|x - y\|_2^2}{\beta}, \qquad (33)$$

where $k$ is a Gaussian kernel with parameters $\kappa, \beta \in \mathbb{R}$. For each $t$, the map $x_0 \mapsto x_t$ is a diffeomorphism, ensuring

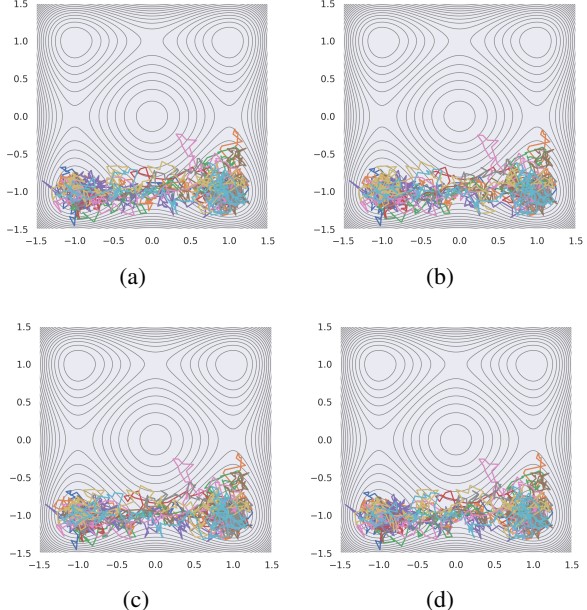

(a)  (b)

(c)  (d)

Figure 5: The first and second coordinates of a 10-dimensional process sampled from the SDE in (30) are shown in (a). The process is conditioned on transitioning from 1 to $-1$ in the first coordinate while remaining at 1 in all others. Panels (b), (c), and (d) depict the paths generated by the BEL-first, BEL-average, and BEL-optimal algorithms, respectively. All plots inclucde underlying contour lines representing the level sets of the potential.

that nearby points remain highly correlated. This property makes learning bridges particularly challenging. An example trajectory of the unconditioned process is shown in Appendix C.2.

We apply our method, BEL average (i.e. $\alpha_t = t$), to learn bridges of the shape SDE, conditioning the process on a circle of radius 1.5. We compare our approach to the methods of Heng et al. (2022) and Baker et al. (2025). The latter uses "adjoint" processes derived from a Feynman-Kac representation for the conditional expectation in (11), while Heng et al. (2022) estimate the time-reversed bridge process by regressing against the time-reversed diffusion bridge drift for small time steps, following Lemma 3.3. However, due to the time-reversal their drift term involves the gradient $\nabla_{x_t} p_{t|s}(X_t = x_t \mid X_s = x_s)$, where differentiation is taken with respect to the final time point rather than the initial one.

Our findings are given in Table 1 and show that BEL average outperforms competing methods. We plot a trajectory from the conditioned process learned via BEL average, alongside a trajectory from the unconditioned process in Appendix C.2. The next best method is Heng et al. (2022), which can be

| Method | Dist |
|---|---|
| BEL average | 0.085 |
| Time reversal | 0.090 |
| Adjoint paths | 0.498 |
| Untrained | 1.396 |

Table 1: A comparison between different methods for learning bridges for shape processes. We see that our proposed method BEL average outperforms other existing methods.

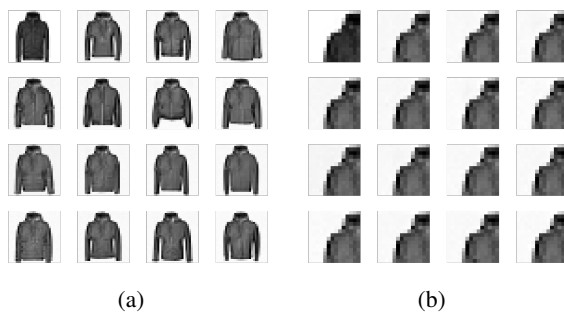

(a)                    (b)

Figure 6: In panel (a), the top-left image shows the ground truth used for conditioning. The remaining 15 images are samples generated by the diffusion model conditioned on the upper-left quarter of the ground truth image. Panel (b) displays only the conditioning inputs: again, the top-left image is the ground truth, while the others show the corresponding conditioned quarters used for generation.

interpreted as first applying BEL to the time-reversed process, approximating the transition densities via Gaussians ((24a)), and using a fixed start point instead of amortisation. However, since their approach involves time-reversal, they do not take gradients of the drift or diffusion terms. BEL average not only uses this gradient information, but non-local information about the score.

### 4.3. Image Experiments

We train a diffusion model using a flow matching loss with a U-Net architecture on the Fashion-MNIST dataset (Xiao et al., 2017).

*Remark* 4.1. The model is trained deterministically using the flow matching loss. We then apply the memoryless schedule from Domingo-Enrich et al. (2024a) to reinterpret the trained model as an equivalent SDE (or diffusion model).

Next, we condition the resulting SDE to produce images with a specified upper-left corner. Importantly, this conditioning does not require adding artificial noise to the observation. More specifically, the law of $X_T$ is by design the data distribution. We condition on $Y = G(X_T)$ where G selects the upper left corner of the image. Once trained we can therefore choose an arbitrary upper left corner $y$ and sample from images matching this, using the learned additional drift $u(X_t, y)$. Note we only need train once and then can sample using any corner. We demonstrate the result by conditioning on the upper-left corner of a jacket image in Figure 6.

*Remark* 4.2. When both the forward and reverse dynamics of an SDE are known, one can construct bridges, as shown in Heng et al. (2022). Diffusion models form a special case of this setting: not only are both directions available, but the reverse dynamics are particularly easy to simulate. This property has been leveraged in Denker et al. (2024) to simplify the methodology of Heng et al. (2022). However, both approaches implicitly assume that the learned score is exact. In contrast, our method makes no such assumption. Since we do not require access to the reverse dynamics, we directly learn the correct conditioning for the learned score—even when it is inexact.

## 5. Conclusion

In this work, we introduced a novel class of loss functions to estimate the control of a conditioned diffusion process. When applied to diffusion models, we discover novel methods to add controls to an already trained diffusion model. As a byproduct, we also find novel denoising score matching objectives, which can be used for time reversal of SDEs, or to train a diffusion model. Our approach leverages Malliavin calculus and integration by parts to handle singular losses, enabling a more stable and accurate estimation. Thanks to its generality, the framework accommodates a wide range of conditioning types—continuous, discrete, noisy, or noiseless—and and could be extended to manifold-valued and infinite-dimensional settings in future works.

### Acknowledgments

EB is supported by the Novo Nordisk Foundation grant NNF18OC0052000, a research grant (VIL40582) from VIL-LUM FONDEN, and UCPH Data+ Strategy 2023 funds for interdisciplinary research. GD and JP were supported by the Engineering and Physical Sciences Research Council [grant number EP/Y018273/1].

### Impact Statement

This work has connections to diffusion models, and could be used in that setting. As such, it shares the societal and ethical implications of generative modelling.

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

# A. Adjoint SDE for Calculating $\mathcal{S}_s$

---

**Algorithm 2** Adjoint SDE Method for Calculating $\mathcal{S}_s$

---

**Require:** Simulation path $\{X_t\}$, Brownian increments $\{\delta B_t\}$, where $t \in \{0, \delta t, \ldots, T\}$.

 1: Initialise $\tilde{\mathcal{S}}_T = 0$.
 2: **for** $t = T - \delta t$ to 0 (backwards in steps of $\delta t$) **do**
 3:    Calculate $\mathcal{D}_t$ via (36).
 4:    Update $\tilde{\mathcal{S}}_t = \tilde{\mathcal{S}}_{t+\delta t} + \mathcal{D}_t$.
 5: **end for**
 6: Compute score process $\mathcal{S}_s$ for $s \in \{0, \delta t, \ldots, T\}$ via (37).

---

We employ an adjoint stochastic differential equation (SDE) method, inspired by adjoint ordinary differential equations (ODEs) (Kidger, 2022; Pontryagin, 2018; Chen et al., 2018), to compute $\mathcal{S}_s$. Specifically, we define the auxiliary variable $\tilde{\mathcal{S}}_s$ as:

$$\tilde{\mathcal{S}}_s := \int_s^T J_{t|s}^\top (\sigma_t(X_t)^\top)^{-1} \, \mathrm{d}B_t, \tag{34}$$

which corresponds to the score process in equation (8) when $\alpha' \equiv 1$, disregarding the normalisation factor $A$.

We assume access to the following data obtained from a discretised Euler-Maruyama simulation of the SDE (6):

- $X_0, X_{\delta t}, \ldots, X_T$: The states of the discretised SDE.

- $\delta B_0, \delta B_{\delta t}, \ldots, \delta B_{T-\delta t}$: The Brownian increments used in the simulation, where $X_{t+\delta t} = X_t + \delta t b_t(X_t) + \sigma(X_t)\delta B_t$.

Note that $\tilde{\mathcal{S}}_T = 0$. We can recursively compute $\tilde{\mathcal{S}}_s$ from $\tilde{\mathcal{S}}_{s+\delta t}$ using:

$$\tilde{\mathcal{S}}_s = \int_s^T J_{t|s}^\top \left(\sigma_t^\top\right)^{-1} (X_t) \, \mathrm{d}B_t = J_{s+\delta t|s}^\top \tilde{\mathcal{S}}_{s+\delta t} + \int_s^{s+\delta t} J_{t|s}^\top \left(\sigma_t^\top\right)^{-1} (X_t) \, \mathrm{d}B_t,$$

making use of the semigroup property $J_{t|s} = J_{t|s+\delta t}J_{s+\delta t|s}$.

Approximating the integral term, we obtain

$$\tilde{\mathcal{S}}_s \approx J_{s+\delta t|s}^\top \tilde{\mathcal{S}}_{s+\delta t} + J_{\tau|s}^\top \left(\sigma_t^\top\right)^{-1} (X_s)\delta B_s, \tag{35}$$

where $\tau$ can be any number in $\tau \in [s, s + \delta t]$. For $\tau = s$, we have $J_{s|s} = \mathrm{Id}$. The term $J_{s+\delta t|s}$ can be approximated using an Euler-Maryuama step on (7), leading to $J_{s+\delta t|s} \approx \mathrm{Id} + \delta t \nabla b_s(X_s) + \nabla \sigma_s(X_s)\delta B_s$.

*Remark* A.1. Here, the term $\nabla \sigma_t(X_t) \, \delta B_t$ is meant as the derivative with respect to $x$ of the map

$$x \in \mathbb{R}^n \mapsto \sigma_t(x)\delta B_t \in \mathbb{R}^n.$$

This allows us to compute the difference term:

$$\mathcal{D}_s := \tilde{\mathcal{S}}_s - \tilde{\mathcal{S}}_{s+\delta t} = \int_s^{s+\delta t} J_{t|s}^\top(\sigma_t(X_t)^\top)^{-1} \, \mathrm{d}B_t \approx (\delta t \nabla b(X_s)^\top + \nabla \sigma(X_s)^\top \delta B_s)\tilde{\mathcal{S}}_{s+\delta t} + (\sigma_t^\top)^{-1}(X_s)\delta B_s. \tag{36}$$

Crucially, note that these Jacobian-vector products are efficiently computed using reverse-mode autodifferentiation, avoiding explicit Jacobian computation.

The algorithm proceeds by initialising $\tilde{\mathcal{S}}_T = 0$ and iteratively computing $\tilde{\mathcal{S}}_t$ backward in time for $t \in \{T - \delta t, \ldots, \delta t, 0\}$. Finally, $\mathcal{S}_s$ is obtained as

$$\mathcal{S}_s \approx A_{T|s}^{-1} \sum_{\substack{t=s,s+\delta t,\cdots}}^{T-\delta t} \alpha_t' \mathcal{D}_t, \tag{37}$$

restoring the normalisation that was dropped in (34).

# B. Denoising Score Matching and Tweedies Formula

Here we show how our results can be applied to derive a formula for the score $\nabla \log p_t(x)$ of a diffusion process. This could be done by applying Proposition 2.4 to a trivial conditioning $Y = 0$ and treating the reverse dynamics of the SDE (6). However, it is instructive to rederive the analogous equation in this simplified setting, which we do in this section. We will see that it generalises Tweedie's formula.

Assume we have an SDE

$$\mathrm{d}X_t = b_t(X_t)\,\mathrm{d}t + \sigma_t(X_t)\,\mathrm{d}B_t, \tag{38}$$

and denote by $p_t$ the density of $X_t$. Then we have the following representation and score matching objective:

**Lemma B.1.** *The score can be represented as*

$$\nabla \log p_t(x) = \mathbb{E}\left[\int_0^t (\sigma_s(X_s)^{-1}J_{t|s}^{-1})^\top \alpha_s'\,\mathrm{d}B_s \mid X_t = x\right].$$

*Here $\alpha_s'$ is a function which satisfies $\int_0^t \alpha_s'\,\mathrm{d}s = 1$, and $J_{t|s}$ is the Jacobian which follows the flow (7).*

*In particular, for any such $\alpha$, the loss*

$$\mathcal{L}(u) = \mathbb{E}\left[\|u_t(X_t) - \int_0^t (\sigma_s(X_s)^{-1}J_{t|s}^{-1})^\top \alpha_s'\,\mathrm{d}B_s\|^2\right] \tag{39}$$

*has a unique minimiser given by $u_t(x) = \nabla \log p_t(x)$.*

*Remark* B.2. The score matching loss (39) is a generalisation of (42) below to nonlinear SDEs. Note that in contrast to (42), the construction in (39) straightforwardly generalises to Riemannian manifolds by replacing the squared norm and the Brownian motion by their Riemannian counterparts (Hsu, 2002), and noting that $J_{t|s}$ maps between the tangent spaces $T_{X_s}M$ and $T_{X_t}M$ in such a way that the stochastic integral is well defined. The loss (39) also generalises to infinite dimensions, replacing the squared norm by a Hilbert space norm and the Brownian motion by an appropriate Wiener process. The relationship between the Wiener process and the chosen norm are however more intricate. In the diffusion model case (the linear case), they have been worked out in Pidstrigach et al. (2024, Section 2.4, Section 6).

*Proof.* By the chain rule for Malliavin derivatives we can write

$$D_s\varphi(X_t) = \nabla\varphi(X_t)J_{t|s}\sigma_s(X_s),$$

where

$$J_{t|s} = \nabla_{X_s}X_t$$

is the Jacobian. This implies that

$$\nabla\varphi(X_t) = D_s\varphi(X_t)\sigma_s(X_s)^{-1}J_{t|s}^{-1},$$

assuming that $\sigma_s$ is invertible. As this relationship holds for all $s$ we can integrate over $s$ to get

$$\nabla\varphi(X_t) = \int_0^t D_s\varphi(X_t)\sigma_s(X_s)^{-1}J_{t|s}^{-1}\alpha_s'\,\mathrm{d}s$$

making use of the fact that $\alpha_s'$ integrates to 1. Taking expectations and using Malliavin integration by parts (see (15)), we arrive at

$$\mathbb{E}[\nabla\varphi(X_t)] = \mathbb{E}\left[\int_0^t D_s\varphi(X_t)\sigma_s(X_s)^{-1}J_{t|s}^{-1}\alpha_s'\,\mathrm{d}s\right] = \mathbb{E}\left[\varphi(X_t)\int_0^t (\sigma_s(X_s)^{-1}J_{t|s}^{-1})^\top \alpha_s'\,\mathrm{d}B_s\right].$$

Based on the framework from Watanabe (1987), we apply this to $\varphi = \delta_x$ the Dirac delta centred at $x \in \mathbb{R}^n$, and get

$$\nabla \log p_t(x) = \nabla_x \log \int p_t(z)\delta_x(z)\,\mathrm{d}z = \frac{1}{p_t(x)}\nabla_x \int p_t(z)\delta_x(z)\,\mathrm{d}z = \frac{1}{p_t(x)}\nabla_x\mathbb{E}[\delta_x(X_t)]$$

$$= \frac{1}{p_t(x)}\mathbb{E}\left[\delta_x(X_t)\int_0^t (\sigma_s(X_s)^{-1}J_{t|s}^{-1})^\top \alpha_s'\,\mathrm{d}B_s\right] = \mathbb{E}\left[\int_0^t (\sigma_s(X_s)^{-1}J_{t|s}^{-1})^\top \alpha_s'\,\mathrm{d}B_s \mid X_t = x\right].$$

Since the conditional expectation is the minimiser of the $L^2$ distance, (39) follows. $\qquad\square$

**Lemma B.3.** *Assume that the forward SDE* (38) *is an Ornstein-Uhlenbeck process*

$$\mathrm{d}X_t = -\frac{1}{2} X_t \, \mathrm{d}t + \mathrm{d}B_t. \tag{40}$$

*Then for the choice* $\alpha'_s = e^{s-t}$ *we get Tweedie's formula:*

$$\nabla \log p_t(x) = \frac{1}{1 - e^{-t}} \mathbb{E}\left[X_t - e^{-t/2} X_0 \mid X_t = x\right], \tag{41}$$

*and, in particular, the loss* (39) *simplifies to the well-known denoising score matching loss:*

$$\mathcal{L}(u) = \mathbb{E}\left[\left\| u(X_t) - \frac{1}{1 - e^{-t}} (X_t - e^{-t/2} X_0) \right\|^2\right]. \tag{42}$$

*Proof.* We have that

$$J_{t|s} = e^{-(t-s)/2} \mathrm{Id}.$$

Therefore,

$$\nabla \log p_t(x) = \mathbb{E}\left[\int_0^t (\sigma(X_s)^{-1} J_{t|s}^{-1})^\top \alpha'_s \, \mathrm{d}B_s \mid X_t = x\right] = \mathbb{E}\left[\int_0^t e^{\frac{t-s}{2}} \alpha'_s \, \mathrm{d}B_s \mid X_t = x\right].$$

If we choose $\alpha_s = e^{s-t}$, then

$$\nabla \log p_t(x) = \frac{1}{1 - e^{-t}} \mathbb{E}\left[\int_0^t e^{-\frac{t-s}{2}} \, \mathrm{d}B_s \mid X_t = x\right]. \tag{43}$$

However, the solution of (40) is given by

$$X_t = e^{-t/2} X_0 + \int_0^t e^{-\frac{t-s}{2}} \, \mathrm{d}B_t.$$

Plugging this into (43), we get

$$\nabla \log p_t(x) = \frac{1}{1 - e^{-t}} \mathbb{E}[X_t - e^{-t/2} X_0 \mid X_t = x].$$

Now, (42) follows again since the conditional expectation is the minimiser of the $L^2$ distance.

$\square$

## C. Experimental Details

### C.1. Controlled Environment Experiments

#### C.1.1. EXPERIMENT SETUP

We discretised the time domain $[0, 1]$ into 200 equivariant grid points for the simulation and used an Euler-Maryuama scheme to simulate paths of the unconditioned SDE (6). We approximated $\mathcal{S}_s$ (8) by starting at the final increment of $\mathrm{d}B_t$ and then using an efficient adjoint method for SDEs to propagate the derivative information backwards. The full Jacobian matrix $J_{s|t}$ is never calculated.

For the algorithms BEL first and BEL last we used $\delta t = \frac{1}{200}$ (see Section 3.3).

We used a batch size of 2048 and iterated through 20 000 batches. We used the Adam optimizer and a neural network architecture which is loosely inspired by UNets. It projects the input data up to 256 dimensions and then has fully connected layers of size $[256, 128, 64, 32, 64, 128, 256]$ with skip connections. The last layer is then a fully connected layer to the output dimension.

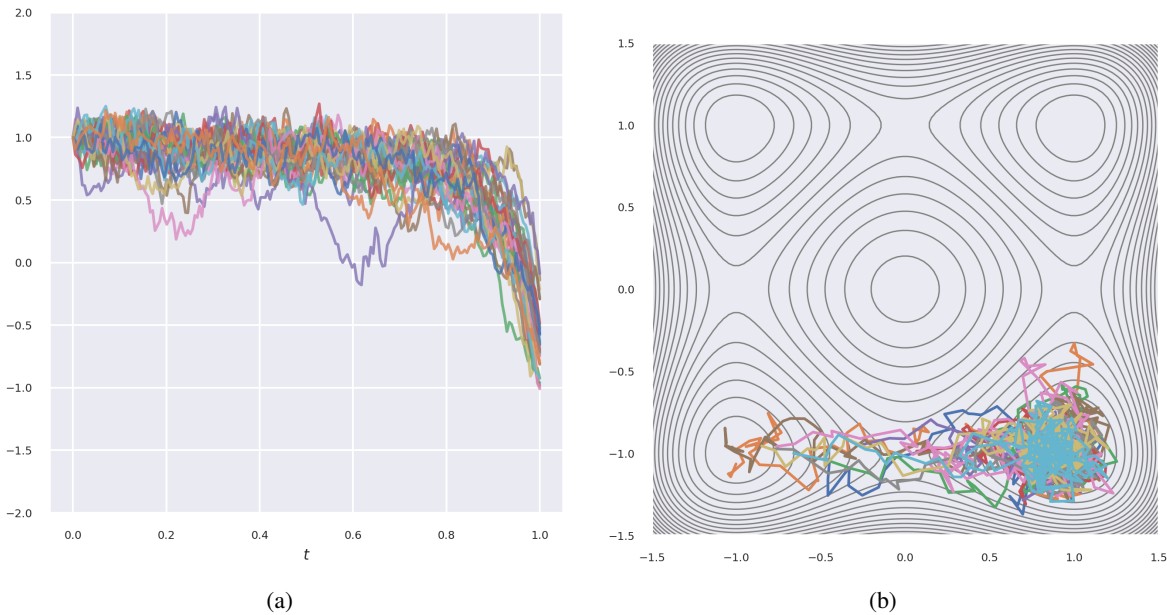

(a)                (b)

Figure 7: We plot the results from the double well experiments for the reparametrisation trick corresponding to (a) Figure 4, the one double well experiment and (b) Figure 5, the ten dimensional double well experiment.

### C.1.2. METRICS

Both metrics compare the simulated paths with paths from the ground truth when started in $y_{\text{init}}$ and conditioned on landing in $y_{\text{final}}$ (see Section 4.1.1), to see if the algorithms can approximate rare events even though they are trained on an amortised objective. We used $15\,000$ simulations of the trained models to calculate the metrics.

**Dist.** This metric calculates the average Euclidean distance of the final state of the path to the conditioned $y_{\text{final}}$.

**MV.** This metric calculates the coordinate-wise mean and variance along the paths of the SDE. It then compares those to the coordinate-wise mean and variance of paths $m^{\text{gen}}, v^{\text{gen}}$ simulated with the ground truth drift, and calculates

$$\text{MV} = \sqrt{\frac{1}{200} \sum_{i=1}^{200} \|m_i^{\text{gen}} - m_i\|^2 + \|(v_t^{\text{gen}})^{1/2} - v_t^{1/2}\|^2}, \tag{44}$$

where $m_i$ and $v_i$ are the variance vectors at time $t = \frac{i}{200}$. The form of (44) is inspired by the Wasserstein-2 distance of two normal distributions.

### C.1.3. RESULTS

Here we provide the tables with the results for our controlled environment experiments for Brownian motion Table 4.

We also provide figures for the reparametrisation trick for the double well experiments in Figure 7.

### C.2. Shape Processes

For the kernel parameters in the SDE (32) we set $\kappa = 0.1$ and $\beta = 1.0$. For all methods we use the neural network and associated parameters in Yang et al. (2025) to train the model with the Adam optimiser. We discretise the time domain $[0, 1]$ into 100 equivariant grid points and use the Euler-Maruyama scheme to simulate paths of the SDEs. For each method, we train on a total of $102.400$ trajectories with a batch size of 128. We compare to the time-reversal method (Heng et al., 2022) and the adjoint method (Baker et al., 2025) using the code provided by Baker et al. (2025).

To evaluate performance, we use the mean pointwise distance between the target shape $\{y_i\}_{i=1}^N$ and the final points of $M$

Table 2: Dimension 1

| Loss | MV | Dist |
|------|-----|------|
| BEL average | $7.6 \times 10^{-2}$ | $3.2 \times 10^{-3}$ |
| BEL first | $8.6 \times 10^{-2}$ | $2.1 \times 10^{-2}$ |
| BEL last | $1.5 \times 10^{-1}$ | $5.6 \times 10^{-2}$ |
| BEL optimal | $1.1 \times 10^{-1}$ | $3.4 \times 10^{-3}$ |
| Reparametrization Trick | $7.6 \times 10^{-2}$ | $2.2 \times 10^{-3}$ |

Table 3: Dimension 10

| Loss | MV | Dist |
|------|-----|------|
| BEL average | $4.5 \times 10^{-1}$ | $2.3 \times 10^{-2}$ |
| BEL first | $4.5 \times 10^{-1}$ | $7.1 \times 10^{-2}$ |
| BEL last | $4.8 \times 10^{-1}$ | $1.9 \times 10^{-1}$ |
| BEL optimal | $4.5 \times 10^{-1}$ | $6.1 \times 10^{-2}$ |
| Reparametrization Trick | $4.5 \times 10^{-1}$ | $9.4 \times 10^{-2}$ |

Table 4: Performance of various algorithms for conditioning a Brownian motion (see Section 4.1.2).

Table 5: Dimension 1

| Loss | MV | Dist |
|------|-----|------|
| BEL average | $8.1 \times 10^{-1}$ | $1.6 \times 10^{-1}$ |
| BEL first | $2.9 \times 10^{-1}$ | $8.4 \times 10^{-2}$ |
| BEL last | $1.1 \times 10^{0}$ | $1.1 \times 10^{0}$ |
| BEL optimal | $8.3 \times 10^{-1}$ | $1.9 \times 10^{-1}$ |
| Reparametrization Trick | $9.2 \times 10^{-1}$ | $2.1 \times 10^{-1}$ |

Table 6: Dimension 10

| Loss | MV | Dist |
|------|-----|------|
| BEL average | $3.4 \times 10^{-1}$ | $3.0 \times 10^{-1}$ |
| BEL first | $2.1 \times 10^{-1}$ | $3.3 \times 10^{-1}$ |
| BEL last | $5.9 \times 10^{-1}$ | $1.1 \times 10^{0}$ |
| BEL optimal | $3.1 \times 10^{-1}$ | $3.0 \times 10^{-1}$ |
| Reparametrization Trick | $5.5 \times 10^{-1}$ | $5.2 \times 10^{-1}$ |

Table 7: Performance of our proposed algorithms for conditioning the double well SDE (see Section 4.1.3). The best-performing algorithm metrics are marked in red.

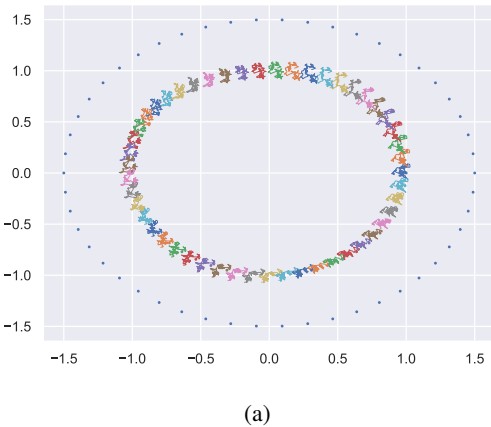
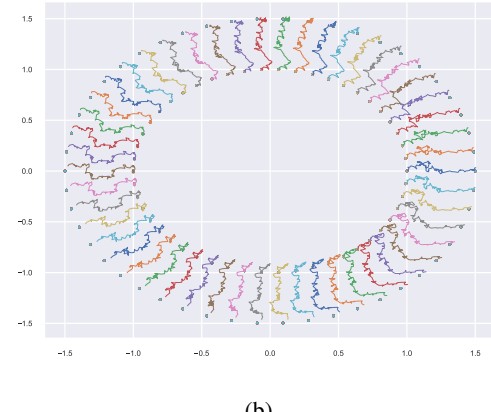

(a)                              (b)

Figure 8: We plot one trajectory from the unconditioned stochastic shape process in (a), started at a circle of radius 1 and one trajectory from the process conditioned via the proposed method BEL average to end at a circle of radius 1.5 in (b).

sampled bridges $\{x^j\}_{j=1}^M$:

$$\frac{1}{MN}\sum_{j=1}^{M}\sum_{i=1}^{N}\|x_i^j - y_i\|_2^2. \tag{45}$$

We compute this metric over $M = 512$ trajectories with $N = 50$ points and report results in Table 1.

## D. Proofs

### D.1. Proof of Proposition 2.4

We start by proving the following preliminary result, which is a slight generalisation of the Bismuth-Elworthy-Li formula (Bismut, 1984; Elworthy & Li, 1994). Results in a similar spirit (although for transition densities instead of scores) can be found in Milstein et al. (2004).

**Theorem D.1.** *Let $\alpha : [0, T] \to \mathbb{R}^{n \times n}$ be a matrix-valued differentiable function such that $A_{T|s} := \alpha_T - \alpha_s$ is invertible for all $s \in [0, T)$. For $T > s$ we have the representation formula*

$$\nabla_{x_s}\mathbb{E}[\varphi(X_T) \mid X_s = x_s] = \mathbb{E}\left[\varphi(X_T)\int_s^T (\sigma_t(X_t)^{-1}J_{t|s}\alpha_t')^\top \mathrm{d}B_t \mid X_s = x_s\right]A_{T|s}^{-1}. \tag{46}$$

*Proof.* **Case $Y = X_T$ (or $G = \mathbf{Id}$).** For any $s < T$ it holds that

$$D_s\varphi(X_T) = \nabla\varphi(X_T)D_sX_T = \nabla\varphi(X_T)J_{T|s}\sigma_s(X_s)$$

by the chain rule of Malliavin calculus (Nualart, 2006, Chapter 2). Therefore,

$$\nabla\varphi(X_T) = D_s\varphi(X_T)\sigma_s(X_s)^{-1}J_{T|s}^{-1}, \tag{47}$$

and

$$\nabla_x\varphi(X_T^x) = \nabla\varphi(X_T)J_{T|0} = D_s\varphi(X_T)\sigma_s(X_s)^{-1}J_{T|s}^{-1}J_{T|0} = D_s\varphi(X_T)\sigma_s(X_s)^{-1}J_{s|0}.$$

Since (47) holds for any $s$, we can integrate along $\alpha_s'$:

$$\nabla_x\varphi(X_T^x) = \int_0^T D_s\varphi(X_T)\sigma_s(X_s)^{-1}J_{s|0}\alpha_s'\mathrm{d}s(\alpha_T - \alpha_0)^{-1}.$$

We now apply the Malliavin integration by parts (15) to obtain

$$\mathbb{E}[\nabla_x \varphi(X_T^x)] = \mathbb{E}\left[\int_0^T D_s\varphi(X_T)\sigma_s(X_s)^{-1}J_{s|0}\alpha_s' ds(\alpha_T)^{-1}\right]$$

$$= \mathbb{E}\left[\varphi(X_T)\int_0^T (\sigma_s(X_s)^{-1}J_{s|0}\alpha_s')^\top dB_s\right](\alpha_T - \alpha_0)^{-1}.$$

$\square$

In order to prove Proposition 2.4 (and later Lemma 3.3), we now prove that the score is a martingale, when conditioned on the observation $Y$:

**Lemma D.2.** *Let $s \geqslant t$, then we have that*

$$\nabla_{X_t} \log p(Y \mid X_t) = \mathbb{E}[\nabla_{X_t} \log p_{s|t}(X_s \mid X_t) \mid Y, X_t]. \tag{48}$$

*Proof.* We have

$$\nabla_{x_t} \log p(Y \mid X_t)$$

$$= \frac{1}{p(Y \mid X_t)}\nabla_{x_t}\int p(Y \mid X_T = x_T)p(X_T = x_T \mid X_t)dx_T$$

$$= \frac{1}{p(Y \mid X_t)}\nabla_{x_t}\int p(Y \mid X_T = x_T)p(X_T = x_T \mid X_s = x_s)p(X_s = x_s \mid X_t)dx_s dx_T$$

$$= \frac{1}{p(Y \mid X_t)}\nabla_{x_t}\int p(Y \mid X_T = x_T)p(X_T = x_T \mid X_s = x_s)p(X_s = x_s \mid X_t)\nabla_{X_t}\log p(X_s = x_s \mid X_t)dx_s dx_T$$

$$= \mathbb{E}[\nabla_{X_t} \log p_{s|t}(X_s \mid X_t) \mid X_t, Y].$$

$\square$

Proposition 2.4 can now be proved using $\varphi = \delta_x$ in Theorem D.1. Intuitively, this corresponds to approximating the Dirac delta distribution by a sequence of peaked Gaussians $\varphi_n$, and then taking the limit. On a technical level, extending Theorem D.1 is supported by the framework of Watanabe distributions, see Watanabe (1987, Section 2).

*Proof of Proposition 2.4.* Recall that the main ideas of the proof have already been outlined in the main text. Applying Theorem D.1 with $\varphi = \delta_{x_T}$, for fixed $x_T \in \mathbb{R}^n$, leads to

$$\nabla \log p_{T|s}(X_T = x_T \mid X_s = x_s) = \frac{1}{p_{T|s}(X_T = x_T \mid X_s = x_s)}\nabla_{x_s}\mathbb{E}[\delta_{x_T}(X_T) \mid X_s = x_s]$$

$$= \mathbb{E}\left[\delta_{x_T}(X_T)\int_s^T (\sigma_t(X_t)^{-1}J_{t|s}\alpha_t')^\top dB_t \mid X_s = x_s\right]\frac{A_{T|s}^{-1}}{p_{T|s}(X_T = x_T \mid X_s = x_s)}$$

$$= \mathbb{E}\left[\int_s^T (\sigma_t(X_t)^{-1}J_{t|s}\alpha_t')^\top dB_t \mid X_s = x_s, X_T = x_T\right]A_{T|s}^{-1}.$$

The final result of Proposition 2.4 follows transposing $\alpha$ and transposing the last equation; this is only a cosmetic change so that the result is more in line with algorithmic implementations.

Now, using Lemma D.2 for $t = T$ we get that

$$\nabla \log p(Y = y \mid X_s = x_s) = \mathbb{E}[\nabla_{X_s} \log p_{T|s}(X_T \mid X_s) \mid Y, X_s]$$

$$= \mathbb{E}\left[\mathbb{E}\left[\int_s^T (\sigma_t(X_t)^{-1}J_{t|s}\alpha_t')^\top dB_t \mid X_s, X_T\right] \mid Y, X_s\right]$$

$$= \mathbb{E}\left[\int_s^T (\sigma_t(X_t)^{-1}J_{t|s}\alpha_t')^\top dB_t \mid Y, X_s\right].$$

$\square$

## D.2. Equivalence to KL-Loss

**Lemma D.3.** *Letting* $\mathcal{L}_{\mathrm{local}}^{t,y}(u_t)$ *be defined as in* (16). *Then it can be interpreted in terms of an amortized Kullback-Leibler divergence between measures on path space as follows:*

$$\int_0^T \mathcal{L}_{\mathrm{local}}^t(u_t)\, \mathrm{d}t = \mathbb{E}\left[\mathrm{KL}(\mathbb{P}^y \mid \mathbb{P}^u)\right] + C$$

*Proof.* We have that

$$
\begin{aligned}
\mathcal{L}_{\mathrm{local}}^{t,y}(u_t) &:= \mathbb{E}[\|u(X_t) - \mathcal{S}_t\|^2 \mid Y = y] \\
&= \mathbb{E}[\|u(X_t) - (\mathcal{S}_t - \mathbb{E}[\mathcal{S}_t \mid X_t, Y] + \mathbb{E}[\mathcal{S}_t \mid X_t, Y])\|^2 \mid Y = y] \\
&= \mathbb{E}[\|u(X_t) - \mathbb{E}[\mathcal{S}_t \mid X_t, Y]\|^2 \mid Y = y] + \mathbb{E}[\|\mathcal{S}_t - \mathbb{E}[\mathcal{S}_t \mid X_t, Y]\|^2 \mid Y = y] \\
&= \mathbb{E}[\|u(X_t) - \nabla \log p(Y = y \mid X_t)\|^2 \mid Y = y] + \mathbb{E}[\|\mathcal{S}_t - \mathbb{E}[\mathcal{S}_t \mid X_t, Y]\|^2 \mid Y = y]
\end{aligned}
$$

where we used that $\mathbb{E}[\mathcal{S}_t \mid X_t, X_T]$ is an $L^2$ orthogonal projection of $\mathcal{S}_s$ onto the set of $\sigma(X_t, X_T)$ measurable random variables which $u(X_t)$ is an element of. Integrating over this we obtain

$$\int_0^T \mathcal{L}_{\mathrm{local}}^{t,y}(u_t)\mathrm{d}t = \mathrm{KL}(\mathbb{P}^y \mid \mathbb{P}^u) + \int_0^T \mathbb{E}[\|\mathcal{S}_t - \mathbb{E}[\mathcal{S}_t \mid X_t, X_T]\|^2 \mid X_T = x_T]\mathrm{d}t,$$

where the second term is independent of $u$. Hence, assuming that the second term is integrable, the result follows by taking expectations on both sides. However, note that they have different finite-sample properties. $\square$

## D.3. Reparametrisation Method

The pathwise reparameterisation trick has been introduced by Domingo-Enrich et al. (2024b) in order to derive the stochastic optimal control matching loss. It is the following result:

**Lemma D.4** ((Domingo-Enrich et al., 2024b), Prop. 1)**.** *Let* $X^x = (X_t^x)$ *be the solution of the SDE* $\mathrm{d}X_t^x = b(X_t^x, t)\, \mathrm{d}t + \sigma(t)\, \mathrm{d}B_t$ *with initial condition* $X_0^x = x$. *Assume that* $f : \mathbb{R}^d \times [0, T] \to \mathbb{R}$ *and* $g : \mathbb{R}^d \to \mathbb{R}$ *are differentiable. For each* $t \in [0, T]$, *let* $M_t : [t, T] \to \mathbb{R}^{d \times d}$ *be an arbitrary continuously differentiable function matrix-valued function such that* $M_t(t) = \mathrm{Id}$. *We have that*

$$
\begin{aligned}
&\nabla_x \mathbb{E}\left[ \exp\left( -\int_0^T f(X_s^x, s)\, \mathrm{d}s - g(X_T^x) \right) \right] \\
&= \mathbb{E}\Bigg[ \left( -\int_0^T M_s \nabla_x f(X_s^x, s)\, \mathrm{d}s - M_T \nabla g(X_T^x) + \int_0^T (M_s \nabla_x b(X_s^x, s) - M_s')(\sigma^{-1})^\top(s)\mathrm{d}B_s \right) \\
&\quad \times \exp\left( -\int_0^T f(X_s^x, s)\, \mathrm{d}s - g(X_T^x) \right) \Bigg].
\end{aligned}
\tag{49}
$$

Looking at the proof of this result in Domingo-Enrich et al. (2024b, Subsec. C.2), we observe that when $g$ is not differentiable, if we impose that $M_T = 0$, then (49) still holds. If we additionally set $f \equiv 0$ and $b$ time-independent, we obtain

$$\nabla_x \mathbb{E}\left[ \exp\left( -g(X_T^x) \right) \right] = \mathbb{E}\left[ \left( \int_0^T (M_s \nabla_x b(X_s^x, s) - M_s')(\sigma^{-1})^\top(s)\mathrm{d}B_s \right) \exp\left( -g(X_T^x) \right) \right]. \tag{50}$$

Next, we prove that relying on our approach, we can recover and generalise this result to compute $\nabla_x \mathbb{E}\left[\varphi(X_T^x)\right]$ for $\varphi$ that are not necessarily strictly positive or differentiable:

*Proof.* We apply Theorem D.1 for $\alpha_s = J_{s|0}^{-1} M_s$ and observe that

$$\sigma_s^{-1} J_{s|0} \alpha_s' = \sigma_s^{-1}(J_{s|0}\alpha_s)' - \sigma_s^{-1}(J_{s|0})'\alpha_s = \sigma_s^{-1}M_s' - \sigma^{-1}\nabla b(X_s)J_{s|0}\alpha_s = \sigma_s^{-1}(M_s' - \nabla b(X_s)M_s).$$

Therefore

$$\mathbb{E}[\nabla_x \varphi(X_T^x)] = \mathbb{E}\left[\varphi(X_T) \int_0^T (\sigma_s^{-1} J_{s|0} \alpha_s')^\top \mathrm{d}B_s\right] (\alpha_T)^{-1}$$

$$= \mathbb{E}\left[\varphi(X_T) \int_0^T (\sigma_s^{-1}(M_s' - \nabla b(X_s) M_s))^\top \mathrm{d}B_s\right] (\alpha_T - \alpha_0)^{-1}.$$

Since $M_s$ is chosen in such a way that $M_0 = \mathrm{Id}$ and $M_T = 0$, we have $\alpha_T - \alpha_0 = -\mathrm{Id}$, and the result follows. $\qquad\square$

### D.4. Gaussian Approximations

Now we use Lemma D.2 to show the equivalence of the two losses:

**Lemma D.5.** *(Lemma 3.3) It holds that*

$$\mathbb{E}[\|u_t(X_t, Y) - \nabla_{X_t} \log p(Y \mid X_t)\|^2] = C + \mathbb{E}[\|u_t(X_t, Y) - \nabla_{X_t} \log p_{s|t}(X_s \mid X_t)\|^2]. \tag{51}$$

*Proof.* We have that

$$\mathbb{E}[\|u_t(X_t, Y) - \nabla_{X_t} \log p(Y \mid X_t)\|^2] = \mathbb{E}[\|u_t(X_t, Y) - \mathbb{E}[\nabla_{X_t} \log p_{s|t}(X_s \mid X_t) \mid X_t, Y]\|^2]$$

by Lemma D.2. Since the conditional expectation $\mathbb{E}[\cdot \mid X_t, Y]$ is a orthogonal projection onto the subspace of $\sigma(X_t, Y)$-measurable random variables in $L^2$, and $s(t, X_t, Y)$ is an element of that subspace, we have that

$$\mathbb{E}[\|u_t(X_t, Y) - \nabla_{X_t} \log p(X_s \mid X_t)\|^2]$$
$$= \mathbb{E}[\|u_t(X_t, Y) - \mathbb{E}[\nabla_{X_t} \log p_{s|t}(X_s \mid X_t) \mid X_t, Y]\|^2] + \mathbb{E}[\|\nabla_{X_t} \log p_{s|t}(X_s \mid X_t) - \mathbb{E}[\nabla_{X_t} \log p_{s|t}(X_s \mid X_t) \mid X_t, Y]\|^2]$$
$$= \mathbb{E}[\|u_t(X_t, Y) - \mathbb{E}[\nabla_{X_t} \log p_{s|t}(X_s \mid X_t) \mid X_t, Y]\|^2] + \mathbb{E}[\|\nabla_{X_t} \log p_{s|t}(X_s \mid X_t)\|^2]$$
$$- \mathbb{E}[\|[\nabla_{X_t} \log p_{s|t}(X_s \mid X_t) \mid X_t, Y]\|^2].$$

This proves the statement. $\qquad\square$

Finally, we prove

**Lemma D.6** (Lemma 3.4). *Approximating $p(X_{t+\delta t} \mid X_t)$ by a Gaussian (24a) and regressing against its score is equivalent to choosing $\alpha_s' = 1_{[t,t+\delta t]}$ and approximating the stochastic integral in $\mathcal{S}_s$ (8) as*

$$\int_t^{t+\delta t} J_{s|t} \mathrm{d}B_s \approx J_{t+\delta t|t}(B_{t+\delta t} - B_t), \tag{52}$$

*and furthermore approximating $J_{t+\delta t|t}$ by an Euler-Maryuama step on (7):*

$$J_{t+\delta t|t} \approx \mathrm{Id} + \delta t \nabla b(t, X_t). \tag{53}$$

*Proof.* Since we approximate

$$p(X_{t+\delta t} \mid X_t) \approx \mathcal{N}(X_t + \delta t b_t(X_t), \delta t),$$

we have that

$$\nabla \log p(X_{t+\delta t} \mid X_t) = (\mathrm{Id} + \delta t \nabla b(t, X_t))(X_{t+\delta t} - (X_t + \delta t b(X_t))) = \frac{1}{\delta t}(\mathrm{Id} + \delta t \nabla b(t, X_t))(B_{t+\delta t} - B_t). \tag{54}$$

Since $\delta = \alpha_{t+\delta t} - \alpha_t$, plugging the normalization $\frac{1}{\alpha_{t+\delta t} - \alpha_t} = \frac{1}{\delta}$ into $\mathcal{S}_s$ (8) shows that the two expressions (8) and (54) are the same. $\qquad\square$

# E. Gaussian Analysis of the Variance

In this Section, we prove Lemma 3.1. First we determine the variance of the Monte Carlo estimator:

**Lemma E.1.** *Set $T = 1$, $n = 1$, $b = 0$ and $\sigma = 1$, i.e., $X_t$ is a one-dimensional Brownian motion conditioned on $X_0 = x_0$ and $X_1 = x_1$. Then the variance of the Monte-Carlo estimator*

$$\nabla \log p_{1|0}(B_1 = x \mid B_0 = x_0) \approx \int_0^1 \alpha'_t J_{t|0}^\top (\sigma_t^{-1})^\top \mathrm{d}B_t$$

*is given by*

$$\frac{1}{\alpha_1 - \alpha_0}\left(\int_0^1 (\alpha'_t)^2 \,\mathrm{d}t + \int_0^1 \frac{(\alpha_1 - \alpha_t)^2}{(1-t)^2}\,\mathrm{d}t + (x_1 - x_0)^2\right), \tag{55}$$

*assuming that $\frac{\alpha_1 - \alpha_s}{\sqrt{1-s}} \to 0$ as $s \to 1$.*

*Proof.* Without loss of generality, we assume that $\alpha'_s$ integrates to 1, and we use the notation $x \equiv x_0$ and $y \equiv x_T$. We need the following preliminary facts,

$$\mathbb{E}[B_s \mid B_1 = y, B_0 = x] = \mathbb{E}[W_s - sW_1 + (1-s)x + sy] = (1-s)x + sy$$

$$\mathbb{E}[\nabla \log p(B_1 = x \mid B_s) \mid B_1 = y, B_0 = x] = \mathbb{E}\left[\frac{x - B_s}{1-s} \mid B_1 = y, B_0 = x\right] = (y - x),$$

where $W_s$ is a standard (unconditioned) Brownian motion. The expectation of the Monte Carlo estimator can then be computed as

$$\mathbb{E}\left[\int_0^1 \sigma_s^{-1} J_{s|0}\alpha'_s \mathrm{d}B_s \mid B_1 = y, B_0 = x\right] = \mathbb{E}\left[\int_0^1 \alpha'_s \mathrm{d}B_s \mid B_1 = y, B_0 = x\right]$$

$$= \mathbb{E}\left[\int_0^1 \alpha'_s \mathrm{d}(W_s + \nabla \log p(B_1 = x \mid B_s)\mathrm{d}s) \mid B_1 = y, B_0 = x\right]$$

$$= \mathbb{E}\left[\int_0^1 \alpha'_s \nabla \log p(B_1 = x \mid B_s)\mathrm{d}s \mid B_1 = y, B_0 = x\right]$$

$$= \mathbb{E}\left[\int_0^1 \alpha'_s \nabla \log p(B_1 = x \mid B_s)\mathrm{d}s\right]$$

$$= \int_0^1 (y - x)\alpha'_s \mathrm{d}s = (y - x).$$

For the square of the estimator we obtain

$$\mathbb{E}\left[\left(\int_0^1 \sigma_s^{-1} J_{s|0}\alpha'_s \mathrm{d}B_s\right)^2\right] = \mathbb{E}\left[\left(\int_0^1 \alpha'_s(\mathrm{d}W_s + \nabla_{B_s}\log p(B_1 = x \mid B_s)\mathrm{d}s)\right)^2 \mid B_1 = y\right]$$

$$= \mathbb{E}\left[\left(\int_0^1 \alpha'_s \mathrm{d}W_s\right)^2\right] + \mathbb{E}\left[\int_0^1 \alpha'_s \mathrm{d}W_s \int \alpha'_s(y - x)\mathrm{d}s\right] + \mathbb{E}\left[\left(\int_0^1 \alpha'_s \frac{y - B_s}{1-s}\mathrm{d}s\right)^2\right]$$

$$= \int_0^1 (\alpha'_s)^2 \mathrm{d}s + 0 + \mathbb{E}\left[\left(\int \alpha'_s \frac{y - B_s}{1-s}\mathrm{d}s\right)^2\right].$$

Moreover,

$$\mathbb{E}\left[\left(\int \alpha'_s \frac{y - B_s}{1-s}\mathrm{d}s\right)^2\right] = \mathbb{E}\left[\int_0^1 \int_0^1 \alpha'_s \frac{B_s - y}{1-s}\alpha'_t \frac{B_t - y}{1-t}\mathrm{d}s\,\mathrm{d}t\right]$$

$$= \int_0^1 \int_0^1 \left(\alpha'_s\alpha'_t \frac{\min(s,t) - st}{(1-s)(1-t)} + \frac{\alpha'_s\alpha'_t\mathbb{E}[B_s - y]\mathbb{E}[B_t - y]}{(1-s)(1-t)}\right)\mathrm{d}s\,\mathrm{d}t \tag{56}$$

The first term in (56) can simplified,

$$
\int_0^1 \int_0^1 \alpha_s' \alpha_t' \frac{\min(s,t) - st}{(1-s)(1-t)} = \int_0^1 \int_0^t \alpha_s' \alpha_t' \frac{s - st}{(1-s)(1-t)} \, \mathrm{d}s \, \mathrm{d}t + \int_0^1 \int_t^1 \alpha_s' \alpha_t' \frac{t - st}{(1-s)(1-t)} \, \mathrm{d}s \, \mathrm{d}t
$$

$$
= \int_0^1 \int_0^t \alpha_s' \alpha_t' \frac{s(1-t)}{(1-s)(1-t)} \, \mathrm{d}s \, \mathrm{d}t + \int_0^1 \int_t^1 \alpha_s' \alpha_t' \frac{t(1-s)}{(1-s)(1-t)} \, \mathrm{d}s \, \mathrm{d}t
$$

$$
= \int_0^1 \int_0^t \alpha_s' \alpha_t' \frac{s}{(1-s)} \, \mathrm{d}s \, \mathrm{d}t + \int_0^1 \int_t^1 \alpha_s' \alpha_t' \frac{t}{(1-t)} \, \mathrm{d}s \, \mathrm{d}t
$$

$$
= \int_0^1 \alpha_s' \frac{s}{(1-s)} \int_s^1 \alpha_t' \mathrm{d}t \, \mathrm{d}s + \int_0^1 \alpha_t' \frac{t}{(1-t)} \int_t^1 \alpha_s' \mathrm{d}s \, \mathrm{d}t
$$

$$
= \int_0^1 \alpha_s' \frac{s}{(1-s)} (\alpha_1 - \alpha_s) \, \mathrm{d}s
$$

$$
= 2 \int_0^1 \frac{s}{1-s} u(s)' \, \mathrm{d}s,
$$

where we have defined $u(s) := \frac{1}{2}(\alpha_1 - \alpha_s)^2$. Now we see that

$$
2 \int_0^1 \frac{s}{1-s} u(s)' \, \mathrm{d}s = -2 \left[ u(s) \frac{s}{1-s} \right]_0^1 + 2 \int_0^1 \left( \frac{s}{1-s} \right)' u(s) \, \mathrm{d}s
$$

$$
= 0 + 2 \int_0^1 \frac{1}{(1-s)^2} u(s) \, \mathrm{d}s
$$

$$
= \int_0^1 \frac{(\alpha_1 - \alpha_s)^2}{(1-s)^2} \, \mathrm{d}s,
$$

where we have used the fact that $\alpha_1 - \alpha_s = o\left(\sqrt{1-s}\right)$ by assumption. The second term in (56) simplifies as follows:

$$
\int_0^1 \int_0^1 \frac{\alpha_s' \alpha_t' \mathbb{E}[B_s - y] \mathbb{E}[B_t - y]}{(1-s)(1-t)} \, \mathrm{d}s \, \mathrm{d}t = \int_0^1 \int_0^1 \frac{\alpha_s' \alpha_t' (x-y)((1-s))(x-y)(1-t)}{(1-s)(1-t)} \, \mathrm{d}s \, \mathrm{d}t
$$

$$
= (x-y)^2 \int_0^1 \int_0^1 \alpha_s' \alpha_t' \, \mathrm{d}s \, \mathrm{d}t = (x-y)^2,
$$

so that collecting all the terms yields the claimed result. $\qquad\square$

In the following determine the optimal choice of $\alpha$ in the simplified setting:

**Lemma E.2.** *Assume the setting from Lemma 3.1. Then the optimal $\alpha$ has derivative*

$$
\alpha_s' = \left( \frac{1}{2} \left( 1 + \sqrt{5} \right) \right) (1-s)^{\frac{1}{2}\left(-1+\sqrt{5}\right)}.
$$

*Proof.* We assume without loss of generality that $\alpha_1 = 1$ and $\alpha_0 = 0$. We need to optimise the following term,

$$
\int_0^1 (\alpha_s')^2 \, \mathrm{d}s + \int_0^1 \frac{(\alpha_1 - \alpha_s)^2}{(1-s)^2} \, \mathrm{d}s = \int_0^1 \left( \beta_{1-s}' \right)^2 + \frac{(\beta_{1-s})^2}{(1-s)^2} \, \mathrm{d}s = - \int_0^1 \left( \beta_s' \right)^2 + \frac{(\beta_s)^2}{s^2} \, \mathrm{d}s,
$$

where we have set $\beta_s := \alpha_1 - \alpha_{1-s}$. We define

$$
F(\beta) := \int_0^1 \left( \beta_s' \right)^2 + \frac{(\beta_s)^2}{s^2} \mathrm{d}s,
$$

since the last term in (19) is independent of $\beta_s$. Assuming that $\alpha$ is a minimizer of $F$, it follows that $\frac{\mathrm{d}}{\mathrm{d}\varepsilon} F(\beta + \varepsilon\varphi) = 0$ for any $\varphi$ such that the $\alpha$ defined through $\beta + \varepsilon\varphi$ satisfies the assumptions of Lemma 3.1. In particular, this condition has to be satisfied for any weakly differentiable $\varphi$ such that $\varphi_0 = \varphi_1 = 1$. Given that, we calculate

$$
\frac{\mathrm{d}}{\mathrm{d}\varepsilon} \int_0^1 \left( \left( \beta_s' + \varepsilon\varphi \right)^2 + \frac{(\beta_s + \varepsilon\varphi)^2}{s^2} \right) \mathrm{d}s = 2 \int_0^1 \left( \varphi' \left( \beta_s' + \varepsilon\varphi' \right) + \frac{\varphi(\beta_s + \varepsilon\varphi)}{s^2} \right) \mathrm{d}s.
$$

Therefore, we arrive at

$$0 \overset{!}{=} \int_0^1 \varphi' {\beta_s}' + \frac{\varphi \beta_s}{s^2} \, \mathrm{d}s = \int_0^1 -\varphi {\beta_s}'' + \frac{\varphi \beta_s}{s^2} \, \mathrm{d}s.$$

Since this equation needs to hold for any continuous $\varphi$ with $0$-boundary conditions it follows that

$$\beta_s'' = \frac{\beta_s}{s^2}.$$

Solving this with the boundary conditions $\beta_0 = 0$ and $\beta_1 = 1$ leads to

$$\beta_s = s^{\frac{1}{2}\left(1 + \sqrt{5}\right)},$$
$$\beta_s' = \left( \frac{1}{2} \left( 1 + \sqrt{5} \right) \right) s^{\frac{1}{2}\left(-1 + \sqrt{5}\right)},$$

as claimed. $\qquad \square$

# F. Related Work

Numerous algorithms exist for bridge simulation. For example, Markov chain Monte Carlo-based methods (Delyon & Hu, 2006; Bayer & Schoenmakers, 2013; Bladt & Sørensen, 2014; Schauer et al., 2017) and more recently, machine learning-based works (Heng et al., 2022; Du et al., 2024; Baker et al., 2025; Zhao et al., 2024).

In the context of diffusion models, the most relevant approaches are Zhang et al. (2023); Denker et al. (2024). These methods rely on a fundamental equality in their derivation:

$$\nabla_{X_t} \log p(X_T | X_t) = \nabla_{X_t} \log p(X_t | X_T) + \nabla_{X_t} \log p(X_t) \approx \nabla_{X_t} \log p(X_t | X_T) + s_\theta(X_t).$$

This formulation allows the first term on the right to be computed analytically, while the second term relies on the pre-trained score network from the diffusion process. Although this approach offers computational efficiency, it assumes that the score network accurately approximates $\nabla \log p_t(X_t)$ – the gradient of the distribution from which $X_t$ is sampled. This assumption frequently fails in practice and may become increasingly invalid after network fine-tuning or other modifications. Our approach avoids this limitation by directly learning optimal control for the current drift estimate $s_\theta(X_t)$, regardless of its origin.

Guidance methods share similar objectives of introducing post-hoc control (Chung et al., 2022; Song et al., 2023; Boys et al., 2023). However, these techniques rely on cheap approximations of the optimal control $\nabla \log p_t(Y | X_t)$, producing samples that deviate from the true posterior or conditional distribution. Instead, they generate samples resembling high-likelihood draws from the prior distribution. While adequate for many applications, these methods cannot provide authentic conditional samples from the Bayesian posterior when such precision is required.

There are also two concurrent works on Malliavin calculus and diffusion models. However, they differ in their goals, techniques and results. In Mirafzali et al. (2025) the authors establish a different Malliavin based score formula. In contrast to Lemma B.1, this construction relies on second-order Malliavin derivatives (Mirafzali et al., 2025, Theorem 3.9) and the inverse of the Malliavin matrix. In Greco (2025) the author uses Malliavin calculus to study diffusion models in infinite dimensions. This can be seen as giving justification for Tweedie's formula (see 2.5) in the infinite dimensional case, extending the results from Pidstrigach et al. (2024).

