# OpenReview forum: "Conditioning Diffusions Using Malliavin Calculus"
_ICML.cc/2025/Conference — ICML 2025 poster_

### Official Review · Reviewer_q3Kt · 2025-03-12

**Overall Recommendation:** 2

**Summary:**

In this paper, the authors propose a framework based on Malliavin Calculus to address the robustness issue associated with singular rewards. Their work focuses on the problem of diffusion bridges, a type of diffusion process with fixed endpoints. Using Doob's h-transform and Malliavin Calculus, the authors derive the Generalized Tweedie's formula. This theoretical result serves as the foundation for their proposed BEL algorithm. Additionally, the authors analyze the variance of the network's training target and select an optimal parameter to minimize this variance. Finally, they conduct experiments on simple diffusion processes and shape processes to demonstrate the empirical performance of their algorithm.

**Claims And Evidence:**

I believe the answer is likely no. The authors explicitly state in the paper: 'Unfortunately, the singular nature of such a measure renders most existing approaches unsuitable: gradients are not well-defined, and naive strategies (e.g., approximating the Dirac measure with highly peaked Gaussians) often face numerical instability.' (lines 55–59). The primary goal of the proposed BEL algorithm is to address this issue. Therefore, I think the authors should present experiments demonstrating improvements in stability and performance for tasks such as flow matching [1] and entropic optimal transport [2] (mentioned in lines 49–55) using their method. In my opinion, the experiments conducted on simple toy examples are insufficient to support such a claim.

[1]. Lipman,Y., Chen,R.T., Ben-Hamu,H., Nickel,M., andLe,M. Flow matching for generative modeling. arXiv preprint arXiv:2210.02747, 2022.

[2]. Shi,Y., DeBortoli,V., Campbell,A., and Doucet, A. Diffusion schrodinger bridge matching. Advances in Neural Information Processing Systems, 36, 2024.

**Essential References Not Discussed:**

Nothing.

**Experimental Designs Or Analyses:**

As mentioned earlier, I believe the authors should conduct experiments on high-dimensional and large-scale datasets (e.g., image datasets) to evaluate the empirical performance of the proposed method.

**Methods And Evaluation Criteria:**

I think the methods is quite interesting, but the authors should conduct experiments on High-Dimensional and Large-Scale datasets (e.g. image datasets) to test the empirical performance of the method.

**Other Comments Or Suggestions:**

Nothing.

**Other Strengths And Weaknesses:**

1. **Lack of High-Dimensional and Large-Scale Experiments**
   While I find the idea presented in the paper both interesting and novel, the current experiments are not sufficient to convince me that the BEL algorithm performs well on high-dimensional data or is scalable. These types of experiments are essential, as downstream tasks of diffusion bridges, such as flow matching, are inherently scalable and often involve high-dimensional generative tasks.

2. **Lack of Analysis on Efficiency and Complexity**
   An analysis of the computational efficiency and complexity of the BEL algorithm is also necessary. Specifically, I believe that estimating the score process within the algorithm could be time-consuming, and this aspect warrants further discussion and evaluation.

**Questions For Authors:**

See the sections above.

**Relation To Broader Scientific Literature:**

I believe that the key tool used to derive the algorithm in the paper is Malliavin Calculus. This is a powerful mathematical framework that allows the application of the integration-by-parts formula on path space. The authors leverage this tool to address the problem in this scenario, which is both innovative and novel.

**Theoretical Claims:**

I have checked the proofs of the theorems.

---

> ### Author Rebuttal · Authors · 2025-04-01
>
> Thanks for your review! We're glad you think the method is interesting. We hope we will be able to address your concerns on the scalability and efficiency of our proposed method, which we discuss below!
>
>
> ### Section q3Kt.1: Larger Experiments and Flow-Matching
> Thank you for the great suggestions. Initially, we mainly targeted the case of conditioning diffusion on a single endpoint, since this can be seen as the most singular, and hence in principle most challenging, reward case (a Dirac delta). However, due to the fact that our method works in these extremal cases, it is actually applicable to a wide range of settings. We followed your proposal and applied our methodology to condition a diffusion model trained on images (could be similarly done for flow matching)!
>
> We first trained a diffusion model on Fashion-MNIST and then used our methodology to condition on the upper left corner of an image. You can see the results here:
> - https://postimg.cc/bsbNgMY2
>
> The upper left image of the right grid is the ground truth. We extracted the upper left quarter of that image as conditioning input (seen as the upper left image in the left grid). We then generated samples from the posterior distribution after conditioning on that upper left image. Training the conditioning drift for this particular problem took us about 30-60 minutes on a high-end GPU. Note that after this training time one can condition on *any* upper left corner.
>
> In general, we can condition on any observation of the form $Y = G(X).$ Here, $Y$ can be discrete (classification) or continuous (upscaling, inpainting, ...), and $G$ can be smooth, but also non-differentiable and even non-continuous, since for diffusion bridges we have the extreme case of $G$ being a Dirac delta. We support both deterministic (noise-free) and stochastic (noisy) versions of G; adding artificial noise is not required but is possible to account for uncertain observations depending on the application. Of course, when $G$ has nice properties, like gradients, it might make sense to include these, but due to our lack of assumptions on the reward, our algorithm is quite generally applicable.
>
> We understand that our paper wasn't to clear on this, and will put more emphasis on this point in the updated version. However, nearly nothing has to change, one only needs to replace $x_T$ by a general condition $y$ in the neural network input during training and evaluation, since we never make use of the form of the observation $x_T = Id(x_T)$ in any of our steps.
>
> ### Section q3Kt.2: Schroedinger Bridges
> Indeed, bridges can be used in the calculation of Schroedinger Bridges. Note that if you can solve the static Schroedinger Bridge problem (Section 3.1 of the paper you linked), and one has access to the Bridges of the reference process, one can interpolate the static coupling of the marginals with the bridges to get the dynamic Schroedinger bridge solution. Therefore, if one has already solved for the bridge process, the problem of solving Schroedinger bridges on path space reduces to just finding the optimal coupling of the marginals!
>
> I hope we were able to dispel your concerns about the applicability of our method to different settings and high dimensional problems. Thank you for motivating us to apply the method to a diffusion model, we are very happy with the results and think this showcases the generality of our method well.
>
> ### Section q3Kt.3: Lack of Analysis on Efficiency and Complexity
> We agree that it maybe was not fully clear how costly the evaluation of the score process is. Therefore we updated our paper to be more detailed on the calculation of the score process. We explain the complexity of the algorithm in the answer to reviewer 4Bsi, you can find it by searching for "Section 4Bsi.3" on this page.
>
> We hope this answers all questions. Furthermore, we hope our experiments on problems with very high energy barries (double well) as well as high dimensions (images), showcase the applicability and efficiency of our method for different kinds of problems.
>
>
> ### Summary
> Thanks again for the review. We believe we have managed to resolve the issues you raised, but please let us know if this is not the case! We hope you agree that the changes to the paper of including a high-dimensional image experiment alongside more details on computing the loss and its complexity will improve it!

---

> > ### Comment · Reviewer_q3Kt · 2025-04-08
> >
> > I'm sorry that I have sent a comment which is not visible to authors... I thank the authors for providing a detailed rebuttal, and I apologize for the delay in my response. Your clarifications have addressed some of my concerns; however, my primary concern remains unresolved. Specifically, while the proposed BEL algorithm is capable of simulating the diffusion bridge without being affected by singularities in the target distribution, it requires the trajectory of the diffusion process as input and update the score process along the jacobian process. This seems less efficient compared to existing methods such as Flow Matching [1] or Bridge Matching [2,3].
> >
> > Although these existing methods [1,2,3] may encounter singularity issues, these challenges can often be mitigated through straightforward techniques such as cutoff adjustments and reparameterization of the network (also known as preconditioning). These strategies have proven sufficient for practical tasks, such as image generation. Given this, I am uncertain whether the proposed BEL method offers a significant advantage or is truly necessary for real-world applications. I'm glad to update my score if the concern is properly addressed.
> >
> >
> > [1]. Lipman, Yaron, et al. "Flow matching for generative modeling." ICLR 2023.
> >
> > [2].Zhou, Linqi, et al. "Denoising diffusion bridge models." ICLR 2024.
> >
> > [3]. De Bortoli, Valentin, et al. "Augmented bridge matching." arXiv:2311.06978 (2023).

---

> > > ### Author Response · Authors · 2025-04-08
> > >
> > > Thank you for your thoughtful feedback. We now better understand the source of the confusion. **We have quite a different use case from  score-based diffusion models, bridge matching and flow matching, and these methods cannot be applied in our setting.**
> > >
> > > ## Source of confusion
> > > The misunderstanding seems to stem from the sentence:
> > > “Diffusion bridges … serve as building blocks for downstream tasks such as flow matching.”
> > >
> > > We have removed this sentence in the revised version, as it may misleadingly suggest our method is an alternative to flow matching or diffusion models, or that we think that these methods are flawed. That is not the case—both are powerful tools with broad applications.
> > >
> > > However, our method is not a generative modeling algorithm itself—it modifies a given reference process (whether physical, economic, or from a generative model) to satisfy specific constraints. In the context of generative modeling, it is most closely related to guidance, reward fine-tuning, or conditioning.
> > >
> > > ## Scope of our method
> > > Let us clarify the scope of our method. We consider a general SDE of the form:
> > > $$ dX_t = b_t(X_t),dt + \sigma(X_t),dW_t, $$
> > > and aim to condition it on a final state $X_T = x$. This corresponds to a reward that assigns infinite value to trajectories ending at $x$, and zero elsewhere (and is therefore very singular).
> > >
> > > The underlying SDE can originate from any system: weather models, molecular dynamics, stock prices, or even from generative algorithms such as diffusion models or flow matching. Our method adds a learned drift to the SDE so that the conditioned endpoint is reached, while preserving the original dynamics. The natural competitors for our method are therefore [1, 2]. The method is, for example, useful for transition path sampling in systems with high energy barriers (see Section 4.1.3 in the paper).
> > >
> > > We hope this clears up the confusion and are happy to address any further questions.
> > >
> > > Best regards,\
> > > The authors
> > >
> > > [1] Heng et al. https://arxiv.org/abs/2111.07243
> > >
> > > [2] Baker et al. https://arxiv.org/pdf/2407.15455

---

### Official Review · Reviewer_4Bsi · 2025-03-14

**Overall Recommendation:** 3

**Summary:**

The authors propose a novel solution to tackling the interesting and challenging problem of diffusion bridges conditioned on singular rewards.
To solve this issue, they make use of theory of Malliavin calculus (essentially stochastic calculus of variations from my understanding) to handle the singularities.
Using this theory they proposed a generalization of Tweedie's formula which the conditional score function $\nabla_x \log p_{T|t}(x_T|x)$ is equal to the conditional expectation of their score process.
They then show to configure the variance of the Monte-Carlo estimator of the conditional score function.
Lastly, they use a few toy experiments to illustrate their method.

## Update after rebuttal
I agree with the other reviewers that the limited demonstration of real world applications is a shortcoming so I adjusted my score.

**Claims And Evidence:**

* Can the claim
"recall the connection of diffusion bridges to Doob’s $h$-transform in Section 2.1, in particular the relevance of conditional score functions"
be a contribution?

 The connection between Doob's $h$-transform and diffusion bridges is quite well known, see [1-4]

[1] Shi et al., *Diffusion Schrödinger Bridge Matching*, NeurIPS 2023, https://proceedings.neurips.cc/paper_files/paper/2023/file/c428adf74782c2092d254329b6b02482-Paper-Conference.pdf

[2] Somnath et al., *Aligned Diffusion Schrödinger Bridges*, UAI 2023, https://proceedings.mlr.press/v216/somnath23a/somnath23a.pdf

[3] Du et al., *Doob’s Lagrangian: A Sample-Efficient Variational Approach to Transition Path Sampling*, NeurIPS 2024, https://arxiv.org/pdf/2410.07974

[4] Brekelmans et al., *On Schrödinger Bridge Matching and Expectation Maximization*, Optimal Transport and Machine Learning Workshop at NeurIPS 2023, https://openreview.net/pdf?id=Bd4DTPzOGO

**Essential References Not Discussed:**

N/A

**Experimental Designs Or Analyses:**

From what I read in the main paper the experiments generally make sense, see questions below for any concerns.

**Methods And Evaluation Criteria:**

They make sense.

**Other Comments Or Suggestions:**

* Footnote 1 is incomplete.

Note, I really enjoyed this paper and found it to be quite interesting. I would be happy to improve the score if the questions and concerns I raised are addressed.

**Other Strengths And Weaknesses:**

* The paper is well-written, and despite being a mathematically dense paper I found it be quite easy to read.
* I found the motivations and arguments for the proposed method to be compelling.
* I found the paper to be rigorous and easy to read, standing next to some of the seminal papers in this field in terms of general quality.

**Questions For Authors:**

1. In Equation (6) what does $\alpha\_t'$ denote? I see $\alpha\_t$ defined in the theorem but not $\alpha\_t'$. Is it the derivative wrt $t$?
2, Wouldn't solving equation (5) amount to the continuous-time analog of forward sensitivity equations, *i.e.*, forward-mode autodiff?

   The work on neural SDEs by Kidger and Li solved the adjoint equations in *reverse-time* corresponding to reverse-mode autodiff. For more details see [1, Section 5.1.4.]
3. Related to Q2 wouldn't simulating vector-matrix products be more efficient than matrix-vector products?
4. In Algorithm 1 in line 1 is $B$ and index or Brownian motion? The notation `for i = 1 to B do` seems strange to me when the next line is Sample $\{X,B\}_i$ ...
5. In Equation (17) is it supposed to be $\alpha_t$ or $\alpha_t'$?
6. In the double well experiment, (Figures 3 and 4) shouldn't samples ideally oscillate between the two wells and not just tend from 1 to -1?
7. In Figures 4 and 5 the impact of different variance schemes (choices for $\alpha_t'$ or $\alpha_t$) seems not to matter much.
8. In Sec 4.2 why does the of variance scheme seem to matter more than in Sec 4.1?

[1] Patrick Kidger, *On Neural Differential Equations*, Ph.D. thesis, https://arxiv.org/pdf/2202.02435

**Relation To Broader Scientific Literature:**

In section 3.2 the authors provide an excellent description of how their work connects with prior work. I have no issues with it.

**Theoretical Claims:**

I read the main paper thoroughly and performed a quick read of the proofs in the appendix. I could have missed details in the appendix.

---

> ### Author Rebuttal · Authors · 2025-04-01
>
> Thanks a lot for your positive review! We're very happy to hear that you really enjoyed this paper! And we were especially happy that you found the general quality to be on par with some of the seminal papers in this field.
> We appreciate your detailed comments and questions, and answer them below.
>
> ### Section 4Bsi.1
> _"Can the claim "recall the connection of diffusion bridges to Doob’s -transform in Section 2.1, in particular the relevance of conditional score functions" be a contribution?"_
>
> Indeed, we agree it is well known. Our intention was to also to use this section to provide an outline to the paper, but we see how this could be confusing. To make it clearer, we will write "recall the (well-known) connection of diffusion bridges" and cite the relevant literature. We hope you agree this makes it clearer.
>
> ### Section 4Bsi.2: "Footnote 1 is incomplete."
> You're right -- the footnote currently continues on the next page. We will fix this! Thank you.
>
> ### Section 4Bsi.3: Computation of the Score Process (Q2)
>
> Thanks for the excellent questions. Indeed, the paper currently lacks some detail on how we compute the score estimator, which may have caused confusion. As you noted, vector-Jacobian products are cheaper via reverse-mode autodiff, and adjoint methods avoid computing the full Jacobian—this is exactly what we do. We simulate paths of the reference SDE (1), then compute the integral (6) in reverse, akin to adjoint ODE methods, but adapted for our setting, picking up $dB_t$ terms along the way instead of only backpropagating a final gradient. We present the full algorithm below for $\sigma = 1$ and Euler-Maruyama discretization, and will include a detailed explanation in the paper. Thank you for the helpful references—we will cite them in the revised version.
>
> **Given**:
> - $(X_0, X_{\delta t}, \ldots, X_T)$
> - $(Z_0, Z_{\delta t}, \ldots, Z_T)$, which is the noise which was used to produce $X_{t + \delta t} = X_t + \delta t  b(X_t) + \sqrt{\delta t}~Z_t$
>
> **Output**:
> - $(S_0, S_{\delta t}, \ldots S_T)$, the score process at all times $t$
>
> **Method**:
> _Initialize_: $S_T = 0$
> _Propagate_: $S_t = S_{t+\delta t}^T (I + \delta t \nabla b(X_t))$
> _At the end_: Divide $S_t$ by $T - t$ to normalize it.
>
> We hope it's now clear that since we left-multiply with the Jacobian, we never need the full Jacobian—just one backprop pass per timestep. This yields  $N = T/\delta t$ regression targets $S_t$ for gradient updates.
>
> Note that the Jacobian in (6) is transposed before right-multiplying with $dB_t$, which is equivalent to left-multiplication. We agree this wasn’t clear and will revise it.
>
> Thanks again for the great question—highlighting this has improved the paper. To show scalability to higher dimensions, we’ve added image experiments (see “Section q3Kt.1” in our reply to Reviewer q3Kt).
>
>
> ### Section 4Bsi.4: Choice of $\alpha$ (Q6 and Q7)
> We found the choices "first", "optimal" and "average" all to do quite well on most problems (however, average and first are easier to implement than optimal). Which one worked the best was problem dependent. Note that "optimal" is only theoretically optimal in a specific setting (a forward process which is a Brownian motion). There are two interesting ways to study optimality:
> - Getting a deeper theoretical understanding of optimal $\alpha$ choices for different classes of SDEs.
> - Optimizing $\alpha$ numerically for a given problem.
>
> However, these were out of the scope of the current work, but they are certainly very interesting for future research.
>
> Note that in Section 4.2 we do not compare different $\alpha$ schemes. We just picked the best-performing one for shape spaces (which was "average"), and compare it to other algorithms from other publications. We observe that we outperform them. We did not observe this problem to behave very differently with regard to the choice of $\alpha$ than the other problems.
>
> ### Section Other Questions
> -- 1. -- Indeed, it's the derivative with respect to $t$. We will add this to the theorem statement to make it clear!
>
> -- 3. -- Thanks for bringing this for our attention. We have updated this to be "for $i=1$ to $N$ ...".
>
> -- 4. --  Yes, thanks - Equation (17) should read $\alpha'_t$. We've corrected this!
>
> -- 5. -- That's a very interesting comment. In our setting (diffusion bridges), there should be no oscillation between the wells as the conditioning "pins down" the endpoint; transitioning back and forth is highly unlikely (even more unlikely than transitioning once!) under the original dynamics. Oscillating back and forth would be expected if the dynamics were set up to equilibrate and target a (spatial) distibution of interest, for example in MCMC type sampling algorithms. Potentially our method could be modified to enhance sampling, but since this is quite different, we believe it is better reserved for future work.
>
> Thanks again for the feedback! We hope we managed to answer your questions sufficiently and address your concerns.

---

> > ### Comment · Reviewer_4Bsi · 2025-04-02
> >
> > I thank the authors for the detailed responses. My concerns on clarity were addressed and I believe the paper is a strong addition to the community. I don't hold the same reservations about experimental scope as the work is mostly theoretical. I will update my scores accordingly.

---

> > > ### Author Response · Authors · 2025-04-08
> > >
> > > Dear Reviewer,
> > >
> > > Thank you for the engaging discussion and thoughtful questions regarding our paper. We appreciate your kind words and are very happy to hear that you enjoyed our article.
> > >
> > > Best regards,
> > > The Authors
> > >
> > > Edit: We were happy to see that you had previously changed your score from a 4 to a 5. We noticed you have since changed it back to a 4. Could we kindly ask what the reason for this is? Thanks again for your review, and if there are still any remaining concerns please let us know!

---

### Official Review · Reviewer_WkxC · 2025-03-14

**Overall Recommendation:** 3

**Summary:**

The work develops a technique for learning the control process that
conditions a diffusion on the terminal state given the initial state.
A key advantage of their method is that it is robust to singular
rewards. They test their algorithm on multi-well toy experiments,
and show that their algorithm successfully allows conditioning on
transitions between the wells. They also test on a Shape process
and obtain better performance than previous methods.

## Update after rebuttal

The rebuttal was good, and I appreciate the additional experiment, but it still seems fairly small scale to me so was not enough to convince me to increase my score, as the extent of the importance of the results in the paper is not conveyed to me. However, I am positive about this paper and maintain my vote leaning toward accept.

**Claims And Evidence:**

Yes, the evidence is sufficient for the claims.

**Essential References Not Discussed:**

None that I can think of.

**Experimental Designs Or Analyses:**

I did not check them in too much detail. The plots of the
diffusion paths show that their method can condition on the transitions.
Moreover, the metrics in the tables also show improvement (they considered
metrics of distance to the target location as well as a kind of
error to the ground truth drift).

**Methods And Evaluation Criteria:**

This seems primarily a theoretical paper, so the experiments
seem appropriate. The practical usefulness of the approach
remains a bit unclear as the experiments do not seem to be
real applications.

**Other Comments Or Suggestions:**

The order of the numbers of some Figures/Tables is strange.
For example, Table 1 is introduced last in the main paper, while other
tables like Table 4 are introduced earlier.

**Other Strengths And Weaknesses:**

The paper is well written.

One weakness is that the experiments are fairly toy, though checking
other related published works, it seems this is common in this field,
and the experiments may be more substantial than usual in similar papers.

**Questions For Authors:**

What is the computation time like?
Is this method practical?

**Relation To Broader Scientific Literature:**

Adequately discussed.

**Theoretical Claims:**

I did not check the proofs thoroughly.

---

> ### Author Rebuttal · Authors · 2025-04-01
>
> We really appreciate your positive review and that you think our paper is well written!
>
> ***One weakness is that the experiments are fairly toy***
>
> We based our experiments on setups from the most closely connected literature and are glad you found them more substantial than in similar papers!
>
> We think the included experiments are good tests for our method since
> 1) In the double well experiment, there is a high energy barrier between transitioning between wells, meaning it happens very rarely. Therefore, conditioning to go from one well to another is challenging.
> 2) In the shape models, points close to one another are very correlated making transitioning to another given shape challenging.
>
> However, we agree these are fairly toy. In order to resolve this we have now conducted an experiment on images, which we hope will convince you of the applicability of our method to numerous situations, including high dimensions!
>
> We use the following setup:
>
> 1. As the base SDE we take a pretrained diffusion model SDE trained to go from the noise distribution to the data distribution (for this we have used Fashion MNIST).
>
> 2. We condition the SDE on the upper left corner of the image to take a given specific value. Practically, this corresponds to the task of completing partial images.
>
> 3. We link our results here: https://postimg.cc/bsbNgMY2 \
> The true "left corner" is from the first image in the top left corner. We see all the samples match well, generating only images with the same left upper corner.
>
> Strictly speaking, this experiment goes slightly beyond the considered set-up, as conditioning on a part of the image translates into "pinning down" only some of the coordinates of the endpoint. Indeed the proposed method is very flexible, and applies to conditioning on any observation $Y= G(X_T)$, with no assumptions on the function $G$ (e.g. $G$ need not be differentiable and $Y$ could be discrete or continuous). For more details, please see our answer to Reviewer q3Kt (see Section q3Kt.1). We realise this is not so clear in the paper right now and will include this discussion!
>
> We hope we have managed to address your concern.
>
>
> ***The order of the numbers of some Figures/Tables is strange.***
> Thanks for pointing this out! We've updated this now so that the tables and figures are named in order of appearance.
>
>
> ***What is the computation time like? Is this method practical?***
>
> Training the conditioning drift for the image task took us about 30-60 minutes on a high-end GPU, so yes, this method can potentially be applied in large-scale settings. Note that after this training time one can condition on *any* upper left corner (for example jackets, shoes, etc). Please do let us know if you want anymore information on this!
>
> For a more detailed complexity analysis of the algorithm (including the number of forward and backward passes) please refer to our answer to Reviewer 4Bsi (Section 4Bsi.3).
>
> Thanks again for your review. We hope we have managed to resolve your concerns by including the extra experiment and an analysis on the complexity of the method.

---

### Official Review · Reviewer_vRjP · 2025-03-15

**Overall Recommendation:** 3

**Summary:**

This paper introduces a novel approach to conditioning diffusion processes using Malliavin calculus, enabling stable training of score-based diffusion bridges. By replacing the ordinary derivative by Malliavin derivative, their framework unifies and extends existing diffusion bridge methods. Through controlled experiments on Brownian motion bridges and double-well potentials, the proposed method performs better than the baselines in accurately modeling conditioned diffusions.

**Claims And Evidence:**

Their main claim is this novel formula for conditional scores in denoising score matching in Theorem 2.1.
To the best of my knowledge, the formulation of replacing ordinary derivatives with Malliavin derivatives is new and the authors provide detailed proof for their main theorem upfront.

**Essential References Not Discussed:**

To my best knowledge, close related works are mentioned in the paper.

**Experimental Designs Or Analyses:**

The experiments are done with two toy experiments set up, given the main contribution of the paper is the theorem, the experimental designs might be acceptable.

**Methods And Evaluation Criteria:**

The proposed method is valid for learning the diffusion bridge problem. The evaluations are toy experiments based.

**Other Comments Or Suggestions:**

In section 4.1.3, some of the references to Figure 5 could be Figure 3?

Figure 1 hardly convey the message of the paper, try either include conclusion in the comments or other examples.

I would start by introducing a motivating example as mentioned  by you, "(approximating the Dirac measure by highly peaked Gaussians) often face numerical instability."  Then showcase that your method is able to solve the instability problem.

**Other Strengths And Weaknesses:**

**Strengths**

Experiments on double-well potential and Brownian motion bridges showcase the effectiveness of the method in capturing rare event transitions.

The derivation of a generalized Tweedie’s formula and path-space integration by parts provides a unifying theoretical framework.


**Weaknesses**

The method assumes that the matrix A and diffusion coefficient \sigma(x_t) are always invertible, which may not hold in degenerate or low-noise regimes.

The quality of the writing sill need to be improved i.e. some of the figures are mis-referenced.

Lack of real world experiments conducted leaving applicability beyond theory uncertain.

**Questions For Authors:**

Some of the questions, please check the theorems section and weakness section of the review.

State-of-art  diffusion based models/flow matching models all have the assumption about the highly peaked Gaussians near the t equals to 0, your method seems to circumvent this problem.  With Jacobian calculation and matrix inverse in calculating S, do you think your method  will be a good fit for high-dimensional task like image generation?


Some of the questions, please check the theorems section and weaknesses section of the review.
State-of-the-art diffusion-based models and flow matching models assume highly peaked Gaussians near t=0, but your method appears to circumvent this issue. With Jacobian calculations and matrix inversions involved in computing S, do you think your method is well-suited for high-dimensional tasks like image generation?

**Relation To Broader Scientific Literature:**

This paper builds on prior work in diffusion bridge learning, score-based generative modeling, and stochastic control, extending methods like Doob's h-transform by using Malliavin calculus for stable conditional score estimation.

**Theoretical Claims:**

For theorem 2.1 as \sigma(X_t) becomes singular, wouldn't you have stability issues with its inverse?

There seems to be a implicit circular dependency between S_t and u_t in your proof for showing the u_t is the unique minimizer, could you make a concise proof in your Appendix to show this is not the case?

---

> ### Author Rebuttal · Authors · 2025-04-01
>
> Thanks for your review! We're glad you see our method as unifying and extending diffusion bridge approaches. Below, we address your questions and concerns.
>
> ***For theorem 2.1 as $\sigma(X_t)$ becomes singular, wouldn't you have stability issues with its inverse? & The method assumes that the matrix A and diffusion coefficient $\sigma(X_t)$ are always invertible...***
>
> Thanks for the question!
>
> For $A$: Since $\alpha$ is user-chosen, it can always be selected to make $A$ invertible. All natural choices for $\alpha$ do so.
>
> For $\sigma$: It is helpful to discuss two slightly different cases:
>
> 1) If $\sigma$ is small but positive definite, stability can be partly managed via small step sizes or adjusting $\alpha$. However, small noise makes the bridge problem harder in general (regardless of the method), as the dynamics become rigid; for $\sigma = 0$, the problem is ill-defined.
>
> 2) The diffusion coefficient is sometimes genuinely rank deficient, for example in the underdamped Langevin dynamics,
> $\mathrm{d}q_t = p_t \, \mathrm{d}t,$
>     $\mathrm{d} p_t = - \nabla V(q_t) \, \mathrm{d}t - \gamma p_t \, \mathrm{d}t + \sqrt{2\gamma} \mathrm{d}W_t,$
> of great interest in molecular dynamics (here the diffusion matrix would be  \begin{pmatrix} 0 & 0 \\\\ 0 & \sqrt{2 \gamma}
> \end{pmatrix} with noise only acting on the $p$-variable). In this case (and in similar other cases), it is possible to extend Theorem 2.1 using pseudo-inverses, and the proposed methodology generalises straightforwardly. On a technical level, this extension requires "hypoellipticity" which guarantees smooth transition probabilities of the underlying dynamics. Since underdamped Langevin dynamics is important for applications, we suggest adding a short explanation in the appendix.
>
> ***There seems to be a implicit circular dependency...***
>
> We are a bit unsure exactly which proof you mean, but we assume you mean in Theorem 2.1. To clarify, the proof steps are as follows:
>
> 1. By Doob’s $h$-transform (Proposition 2.2) for eq (8) to coincide with the conditional law, we need to show that $u^*(x; x_T) = \nabla \log p_{T\mid t}(X_T = x_T \mid X_t = x)$.
> 2. By Proposition 2.3, we know $\nabla \log p_{T\mid t}(X_T = x_T \mid X_t = x) = \mathbb{E}[\mathcal{S}_t \mid X_t=x, X_T=x_T]$.
> 3. In the proof of Theorem 2.1 we show that the minimiser of $\mathcal{L}(u)$ is given by $\mathbb{E}[\mathcal{S}_t \mid X_t=x, X_T=x_T]$. Since this is equal to the term arising from Doob’s $h$-transform, eq (8) coincides with the SDE under the conditional law.
>
> In particular, in equations (14) and (15), $\mathcal{S}_t$ is already fixed and does not depend on $u$, hence there is no circularity.
>
> If this is what you mean, we’d be glad to include the above steps in proof of 2.1 making it more explicit!
>
>
> ***The quality of the writing still needs to be improved...***
>
> Thanks for pointing this out! We will fix this immediately.
>
> ***Lack of real world experiments conducted leaving applicability beyond theory uncertain.***
> We based our experiment setup on papers in the literature that also concentrate on similar problems [1, 2, 3, 4, 5]. Indeed, the double well experiment is a challenging problem in conditioning due to the high energy barrier between wells and therefore the transitioning events are very rare. However, we agree it is interesting to consider image generation, and have since experimented with this!
>
> We condition a pretrained model to only generate images matching with a given top left corner. Please see our answer to Reviewer q3Kt (Section q3Kt.1) for details, and see the samples linked here: https://postimg.cc/bsbNgMY2.
>
> We hope we've conveyed the method's applicability and flexibility! We'll include this experiment to show it scales to high-dimensional SDEs and has practical relevance beyond the theory.
>
> [1] https://arxiv.org/abs/2111.07243\
>
> [2] https://arxiv.org/abs/2407.15455\
>
> [3] https://link.springer.com/article/10.1007/s42985-021-00102-x\
>
> [4] https://arxiv.org/pdf/2312.02027v4 \
>
> [5] https://projecteuclid.org/journals/bernoulli/volume-23/issue-4A/Guided-proposals-for-simulating-multi-dimensional-diffusion-bridges/10.3150/16-BEJ833.full.
>
> ***Figure 1 hardly convey the message...start by introducing a motivating example...***
>
>
> We agree we should include more information in this first image, and update it as follows: https://postimg.cc/NK7gq2JB.
> We hope you agree that this better illustrates the motivation and challenge of the paper, and it also gives us the opportunity of referring to it in the introduction when explaining the reward (and its potential singularity).
>
> **...do you think your method will be a good fit for high-dimensional task like image generation?**
>
> Yes, as we have seen from our experiment on images it does scale to high-dimensional tasks! The Jacobian calculation can be done very efficiently using vector-matrix products; for more details on this and on computing $\mathcal{S}$ please see our answer to Reviewer 4Bsi (Section 4Bsi.3).

---

### Decision · Program_Chairs · 2025-05-01

**Decision:**

Accept (poster)

**Comment:**

This is an interesting paper which explores applications of Malliavin calculus to diffusions. The paper is praised by the reviewers for the novelty but also various suggested are made to improve the final version of the paper. I hope the authors will implement some of these in the final version.